# *Cis*-regulatory evolution of the recently expanded *Ly49* gene family

Changxu Fan[1,2], Xiaoyun Xing [1,2], Samuel J. H. Murphy[3,4], Jennifer Poursine-Laurent[5], Heather Schmidt[1,2], Bijal A. Parikh[6], Jeesang Yoon[5], Mayank N. K. Choudhary[1,2], Naresha Saligrama [3,6,7,8,9], Sytse J. Piersma [5,10,12] ✉, Wayne M. Yokoyama [5,6,12] ✉ & Ting Wang [1,2,11,12] ✉

Comparative genomics has revealed the rapid expansion of multiple gene families involved in immunity. Members within each gene family often evolved distinct roles in immunity. However, less is known about the evolution of their epigenome and *cis*-regulation. Here we systematically profile the epigenome of the recently expanded murine *Ly49* gene family that mainly encode either inhibitory or activating surface receptors on natural killer cells. We identify a set of *cis*-regulatory elements (CREs) for activating *Ly49* genes. In addition, we show that in mice, inhibitory and activating *Ly49* genes are regulated by two separate sets of proximal CREs, likely resulting from lineage-specific losses of CRE activity. Furthermore, we find that some *Ly49* genes are cross-regulated by the CREs of other *Ly49* genes, suggesting that the *Ly49* family has begun to evolve a concerted *cis*-regulatory mechanism. Collectively, we demonstrate the different modes of *cis*-regulatory evolution for a rapidly expanding gene family.

During gene duplications, *cis*-regulatory elements (CREs) may be duplicated together with their target genes. Experimental evidence[1,2] has shown that the divergence of paralogous CREs could lead to partitioning of the ancestral expression pattern among paralogs (copies of the ancestral gene generated by duplication), a process known as subfunctionalization, which could contribute to the retention of paralogs in the genome[3]. However, most of these studies focused on paralogs generated by ancient whole genome duplications, and little is known about the evolution of paralogous *cis*-regulation in the context of recent and on-going tandem gene duplications. Importantly, rapid tandem duplications could generate multiple

paralogs clustered in a small genomic space, which could lead to co-regulation of these genes.

The *Ly49* (also known as *Klra*) gene family is one of the fastest evolving murine gene families, existing as a single copy in most mammals including humans, but has recently expanded into a large family in murids[4]. Most murine *Ly49* genes encode natural killer (NK) cell surface receptors that recognize distinct ligands and transduce either inhibitory or activating signals. Inhibitory Ly49 receptors recognize various MHC-I molecules and inhibit killing of host somatic cells[4], whereas some activating Ly49 receptors can directly stimulate NK cell killing of virus-infected cells[5-8]. Ly49H, for example, recognizes

[1]Department of Genetics, Washington University School of Medicine, St. Louis 63110, USA. [2]The Edison Family Center for Genome Sciences & Systems Biology, Washington University School of Medicine, St. Louis 63110, USA. [3]Department of Neurology, Washington University School of Medicine, St. Louis 63110, USA. [4]Medical Scientist Training Program, Washington University School of Medicine, St. Louis 63110, USA. [5]Division of Rheumatology, Department of Medicine, Washington University School of Medicine, St. Louis 63110, USA. [6]Department of Pathology and Immunology, Washington University School of Medicine, St. Louis 63110, USA. [7]Bursky Center for Human Immunology and Immunotherapy Programs, Washington University School of Medicine, St. Louis 63110, USA. [8]Hope Center for Neurological Disorders, Washington University School of Medicine, St. Louis 63110, USA. [9]Center for Brain Immunology and Glia (BIG), Washington University School of Medicine, St. Louis 63110, USA. [10]Siteman Cancer Center, Washington University School of Medicine, St. Louis 63110, USA. [11]McDonnell Genome Institute, Washington University School of Medicine, St. Louis 63110, USA. [12]These authors jointly supervised this work: Sytse J. Piersma, Wayne M. Yokoyama, Ting Wang. ✉e-mail: spiersma@wustl.edu; yokoyama@wustl.edu; twang@wustl.edu

the virus-encoded MHC-I like protein m157 and stimulates NK cell killing[9,10]. Although mainly expressed in NK cells, inhibitory Ly49 receptors are also expressed in certain other immune lineages, including subsets of T cells. By contrast, the expression of activating Ly49 receptors is highly restricted to NK cells[11,12]. In humans, an unrelated gene family, known as *KIR*, has evolved to perform highly similar functions, forming a classic case of convergent evolution[13].

The expression of *Ly49* genes is described as stochastic and variegated[14], i.e., a given *Ly49* gene is only expressed in a subset of otherwise homogenous NK cells and individual NK cells often express more than one *Ly49* gene, even of the same functional type, resulting in overlapping subsets of NK cells with respect to the expression of individual *Ly49* genes. Once turned on, *Ly49* expression is mitotically stable[15]. The expression choice of different *Ly49* genes is considered as mostly independent, but exceptions have also been observed[11,14]. Previous research focusing on two inhibitory *Ly49* genes have revealed the association between *Ly49* expression choice and promoter methylation[16]. In addition, seminal works have identified a conserved CRE upstream of most *Ly49* genes, named *Pro1* or *Hss1*, regulating the expression of some inhibitory *Ly49* genes[17–20]. However, the *cis*-regulation of activating *Ly49* genes have not been characterized. It also remains unknown if and how *Ly49* epigenome and *cis*-regulation have diverged. Furthermore, the proximity of different *Ly49* genes suggests that they could be co-regulated by sharing CREs, which has not been extensively explored. Importantly, *Ly49* genes display extensive interstrain polymorphisms: different mouse strains usually have different sets of *Ly49* genes and alleles[21]. Therefore, in this work, to further elucidate the *cis*-regulation of *Ly49* expression and how it diversified during evolution, we carry out systemic epigenetic mapping of NK cells with different *Ly49* expression status from 3 strains of mice.

## Results

### Ly49 genes feature lowly accessible promoters and constitutively accessible enhancer-like elements

We focused on the *Ly49* clusters in 3 inbred mouse strains with diverse *Ly49* gene compositions, including the C57BL/6J (B6)[22], 129S6/SvEvTac (129)[23], and NOD/ShiLTJ (NOD)[24] strains. We labeled protein-coding *Ly49* genes as inhibitory or activating based on the sequences of their encoded proteins, and annotated pseudogenes as inhibitory or activating based on their sequence similarities to protein-coding *Ly49* genes, according to published annotations[22–24] (Supplementary Fig. 1, Methods).

We first sought to define *Ly49* promoters by experimentally identifying transcription start sites (TSSs). Conflicting prior results have indicated that *Ly49* transcription could initiate from multiple putative CREs conserved across mouse *Ly49* genes, including *Pro1*, *Pro2*, *Pro3*, and transposon RMER5[17,18,25–27]. To clarify their roles, we isolated splenic NK cells using fluorescence-activated cell sorting (FACS, Supplementary Fig. 2), and generated a base pair-level TSS profile using nanoCAGE[28] (Methods, Supplementary Data 1). Using stringent read mapping criteria, we confirmed the canonical TSS located at *Pro2* for all expressed *Ly49* genes, as well as another major TSS located at *Pro3* for a subset of *Ly49* genes, including *Ly49g* of the B6 strain (*B6.Ly49g*)[25] (Supplementary Fig. 3a–d). For *Ly49a/g/o* genes, transcription initiation from *Pro3* is concentrated at the same homologous site (blue arrow in Supplementary Fig. 3b, e) conserved among these genes. Interestingly, this site is not conserved in other *Ly49* genes. The main TSS of *B6.Ly49d* is located downstream of the start codon conserved across most *Ly49* genes. We identified a non-canonical start codon specific to *Ly49d/r* created by a C to T substitution 63 bp downstream of the conserved start codon (Supplementary Fig. 3e), which would likely result in a functionally equivalent form of the Ly49D receptor, as the missing N-terminus amino acid residues do not directly participate in intracellular signaling[4]. Consistent with a recent report[27], we also detected 2 other elements

harboring non-canonical *Ly49* TSSs, i.e., the *Pro1* elements of *B6.Ly49g* and *129.Ly49g*, and the RMER5 transposon upstream of *Ly49i* genes (Supplementary Fig. 4). Together, our data clarified *Ly49* TSS distributions and demonstrated their diversification across different *Ly49* genes.

Next, we profiled *Ly49* CREs using ATAC-seq. Since *Ly49* genes are stochastically expressed in NK cells, we compared the ATAC-seq data from B6 splenic NK cells sorted based on the expression of *B6.Ly49a* and *B6.Ly49h*, representing inhibitory and activating *Ly49* genes, respectively. *Ly49* promoter (defined by nanoCAGE peaks) accessibility correlated with gene expression (Fig. 1a, Supplementary Fig. 5a, b). However, both *B6.Ly49a* and *B6.Ly49h* promoters exhibited low accessibility, even in Ly49A[+] and Ly49H[+] cells, respectively. Generalizing this observation, *Ly49* promoters are markedly less accessible compared with other gene promoters in NK cells (Fig. 1b), consistent with their lack of transcription factor binding (Fig. 1a). Including multimapping reads in the analysis did not change this result (Supplementary Fig. 6). Next, we profiled *Ly49* DNA methylation using whole genome as well as targeted bisulfite sequencing, focusing on the promoters of 2 activating *Ly49* genes, *B6.Ly49d* and *B6.Ly49h*. Our results showed that these promoters are only unmethylated when the corresponding genes are expressed (Fig. 1c, d), consistent with previous results on inhibitory *B6.Ly49a* and *B6.Ly49c*[16]. In addition, H3K4me3, a histone modification marking active promoters, extends from *Ly49* promoters into gene bodies and forms broad domains. The total intensity of H3K4me3 correlated with *Ly49* expression, suggesting its selective deposition at promoters of expressed alleles, and providing a potential mechanism for the maintenance of stable expression (Supplementary Fig. 5c, d)[29,30]. In addition to promoters, we discovered a major ATAC-seq peak (MAP) 5-8 kb upstream of most *Ly49* gene promoters expressed in conventional NK cells, except for *B6.Ly49d* (Fig. 1a, c, Supplementary Fig. 5e). No proximal MAP was found upstream of *B6.Ly49d* even in sorted Ly49D[+] cells. The proximal MAP of *B6.Ly49a* is constitutively accessible in both Ly49A[+] and Ly49A[-] cells (Fig. 1a). Similarly, the proximal MAP of *B6.Ly49h* is constitutively accessible and unmethylated in both Ly49H[+] and Ly49H[-] cells (Fig. 1a, c). In addition, MAP accessibility is not affected by inhibitory Ly49-mediated NK cell licensing (Supplementary Fig. 7a, 8). In general, MAPs show little transcriptional activity, with few exceptions (Fig. 1a, Supplementary Fig. 3). On the other hand, they are clearly enriched for enhancer marks H3K27ac[31] and p300[32], and are bound by NK cell transcription factors T-bet[31] and Runx3[33] (Fig. 1a, Supplementary Fig. 5e), suggesting that they are likely enhancers in mature NK cells. Nonetheless, we did notice low levels of H3K4me3 signals at some MAPs, such as the MAP near *B6.Ly49a* (Fig. 1a), consistent with their reported promoter activities in immature NK cells[17,18]. Together, our data suggest that each *Ly49* gene forms a *cis*-regulatory unit with one or more promoters lowly accessible only when expressed, and a proximal enhancer-like element constitutively accessible in all NK cells.

### Inhibitory and activating Ly49 genes are regulated by two separate sets of CREs

To account for the diversity of *Ly49* genes among mouse strains, we performed additional ATAC-seq on splenic NK cells FACS-isolated from the NOD and 129 strains, which have different *Ly49* haplotypes from B6, and projected *Ly49* ATAC signals across the 3 strains onto a *Ly49* consensus sequence (Methods). *Ly49* chromatin accessibility profiles clearly segregated into 2 major groups, with MAPs occurring at 2 different conserved elements, which we termed *MAP1* and *MAP8*, respectively (Fig. 2a, Supplementary Fig. 9a, b). The same pattern was seen for the binding profiles of known NK cell transcription factors T-bet and Runx3[31,33] (Supplementary Fig. 9c). This segregation directly reflects the functionality of the encoded protein: inhibitory *Ly49* genes are associated with accessible *MAP1s*, whereas activating *Ly49* genes are associated with accessible *MAP8s*.

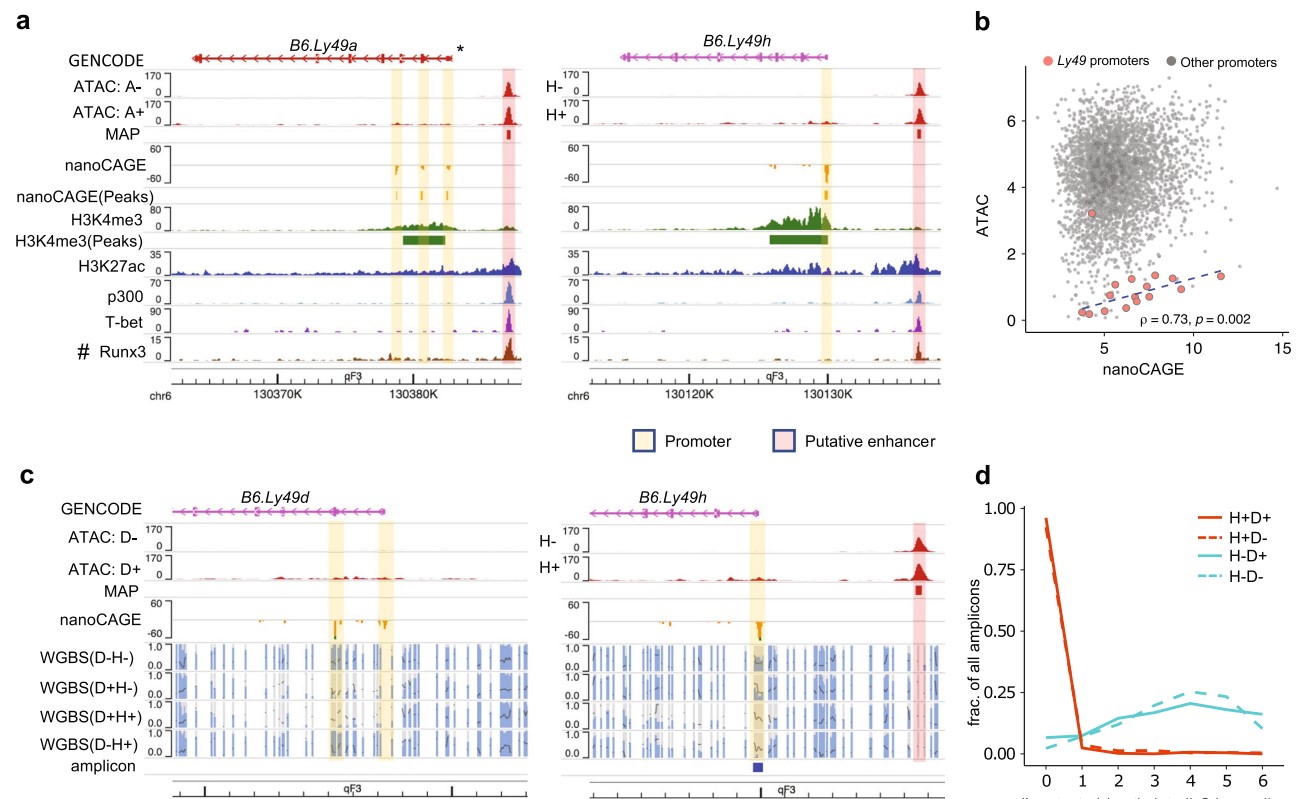

**Fig. 1 | *Ly49* genes feature lowly accessible promoters and constitutively accessible enhancer-like elements. a** The epigenetic signature of *B6.Ly49a* and *B6.Ly49h*. A: Ly49A; H: Ly49H. Sources of public data: H3K4me3: GSM4314407[31]; H3K27ac: GSM4314409[31]; p300: GSM2056372[32]; T-bet: GSM4314405[31]; Runx3: GSM1214531[33]. * indicates manual curation of GENCODE (M19) *B6.Ly49a* annotation. The original GENCODE annotation designated *Pro1* as *B6.Ly49a* promoter, which is not supported by our nanoCAGE data. #: data generated from the ICR mice, but aligned to the B6 genome, due to the lack of a ICR reference genome. **b** Scatter plot of ATAC-seq signals vs nanoCAGE signals for each active promoter (defined by nanoCAGE peaks. Methods). ATAC-seq and nanoCAGE signals are the average of 6 and 4 libraries, respectively. Both axes were normalized and log2 transformed (Methods). Pearson correlation was calculated based on *Ly49* promoters only

($n = 15$ *Ly49* promoters). One of the promoters of *B6.Ly49g* (top red dot) was excluded from the correlation analysis because it overlaps a MAP (Supplementary Fig. 4a). *P*-value was calculated from two-tailed Student's *t* statistic. **c** Whole genome bisulfite sequencing (WGBS) data for sorted Ly49D⁻H⁻, Ly49D⁺H⁻, Ly49D⁺H⁺, and Ly49D⁻H⁺ NK cells, visualized using methylC tracks, where each bar marks a CpG, with gray and blue indicating the proportion of reads where this CpG was unprotected (unmethylated) or protected (methylated), respectively. To achieve allele-level resolution, CB6F1/J animals, which only encode *Ly49d* and *h* on one *Ly49* allele, were used (Methods). D: Ly49D; H: Ly49H. **d** There are 6 CpGs covered by the amplicon-based bisulfite sequencing (indicated in **c**). The plot shows the fraction of amplicons where an indicated number (*x*-axis) of CpGs were protected (methylated). D: Ly49D; H: Ly49H. Source data are provided as a Source Data file.

Previous landmark studies have examined the role of several *MAP1s* under the name of *Pro1* or *Hss1*[17–20]. However, to our knowledge, *MAP8s* have never been described. We selected the proximal *MAP8* of *B6.Ly49h* (*MAP8.B6.Ly49h*) as a representative *MAP8* element for in-depth study, due to the functional importance of *B6.Ly49h* in murine viral defense[9]. We generated knockout mice with *MAP8.B6.Ly49h* removed (*MAP8.B6.Ly49h* KO; Fig. 2b). KO animals showed minimal *B6.Ly49h* expression in both mature and immature NK cells (Fig. 2c–e, Supplementary Fig. 7b, Supplementary Fig. 9d, e), and failed to control murine cytomegalovirus (MCMV) replication in the spleen similar to Ly49H deficient mice[9] (Fig. 2f), underscoring the essentiality of *MAP8* in murine immunity. Considering the similarity among *Ly49* paralogs, we sought to rule out the possibility that off-target CRISPR-editing contributed to the observed phenotype. We assembled the *Ly49* locus of *MAP8.B6.Ly49h* KO mice using targeted nanopore long-read sequencing. The assembled contig showed minimal divergence from the WT B6 *Ly49* locus (Supplementary Fig. 9f). No large-scale structural variation was detected in the assembled contig (Fig. 2g), and the rare residual differences from the WT B6 *Ly49* locus can be attributed to the error of nanopore sequencing at repeats and DNA homopolymers (Supplementary Fig. 9g–i). Therefore, it is unlikely that CRISPR off-target effects contributed to the phenotype. Similarly, ablation of *B6.Ly49a* expression was observed in *MAP1.B6.Ly49a* KO animals from

a previous study[20]. Together, these data strongly suggest that *MAP1s* and *MAP8s* act as enhancers regulating the expression of inhibitory and activating *Ly49* genes, respectively (Fig. 2h). Interestingly, *B6.Ly49d*, despite being highly expressed in NK cells, do not have accessible proximal *MAP1* or *MAP8*, suggesting that not all *Ly49* genes are regulated by proximal *MAP1s or MAP8s*. At the DNA sequence level, *MAP1* and *MAP8* paralogs are both conserved upstream of each *Ly49* gene, with very few exceptions (Fig. 2h, Supplementary Fig. 10, 11). However, the sequences of both *MAP1* and *MAP8* have diverged between inhibitory and activating *Ly49* genes. The binding motif of Runx3, for example, is conserved at the proximal *MAP1s* of inhibitory *Ly49* genes, but has decayed at the proximal *MAP1s* of activating *Ly49* genes (Supplementary Fig. 10), consistent with Runx3 binding profiles (Supplementary Fig. 9c). Moreover, although accessible *MAP1s* and *MAP8s* have similar transcription factor binding profiles (Supplementary Fig. 9c), we observed no regions longer than 35 bp with >= 60% identity between the consensus sequences of *MAP1* and *MAP8*, suggesting that *MAP1* and *MAP8* are not likely recent duplicates of each other (Supplementary Fig. 9j). Overall, our data revealed an evolutionary association between *Ly49* protein function and CRE choice.

Subsets of T cells, such as CD8⁺ regulatory T cells (CD8 T reg cells), are known to express inhibitory Ly49 receptors, but not activating ones[34]. We next investigated if this phenomenon is associated with the

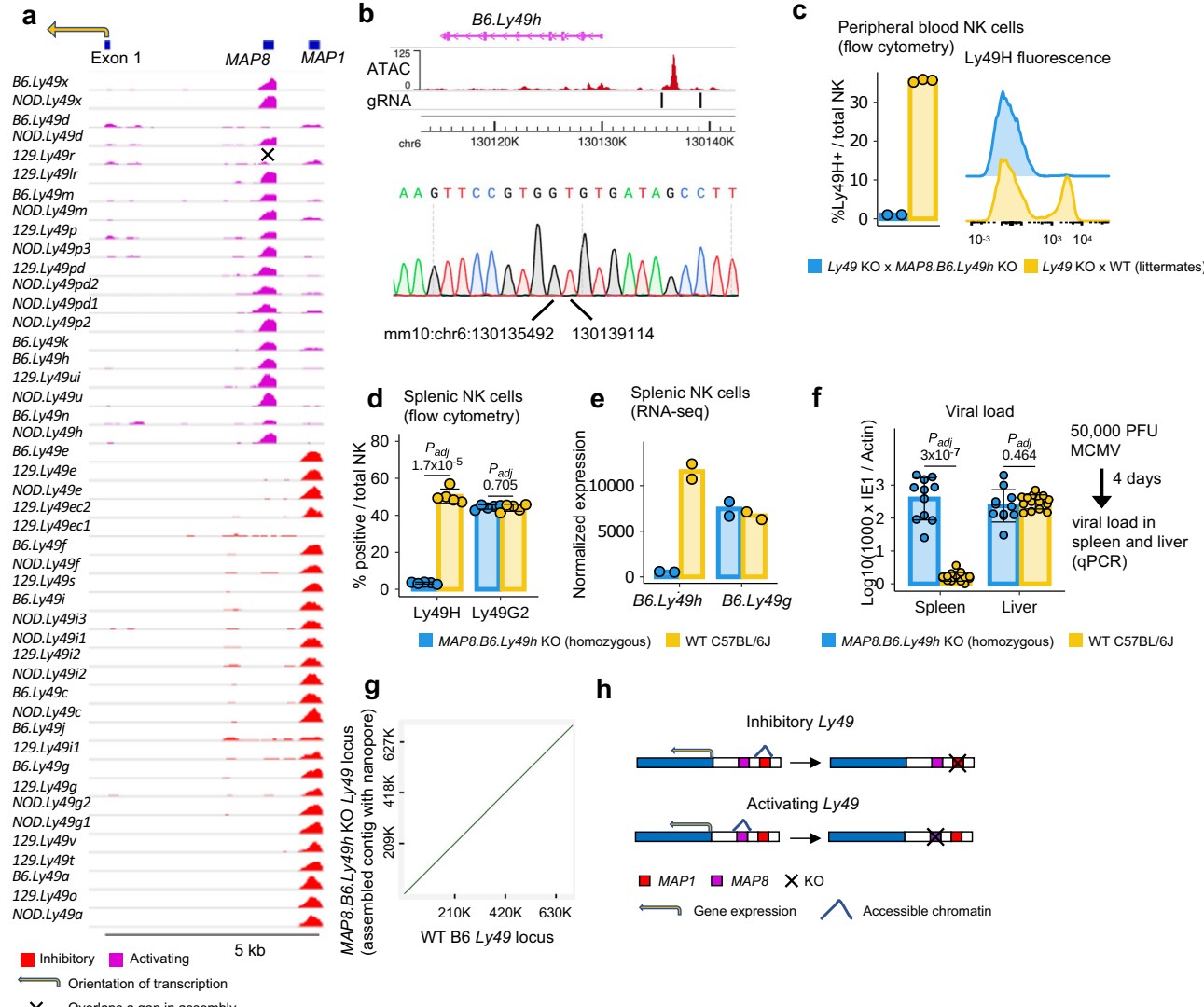

**Fig. 2 | Inhibitory and activating *Ly49* genes are regulated by two separate sets of proximal CREs. a** Sequence alignment-based pileup view of ATAC signals on the coordinates of the *Ly49* consensus sequence (Methods). Only reads with 100 % match to reference were included. No mapping quality (MAPQ) filters were applied. The same plot based on reads passing MAPQ > = 8 is presented in Supplementary Fig. 9a. **b** The exact location of KO for the *MAP8.B6.Ly49h* KO allele. **c** Percentages of Ly49H⁺ cells out of total NK cells in peripheral blood (flow cytometry). The *Ly49* KO allele expresses no *Ly49* genes in the *Ly49* cluster on the protein level (Methods). *n* = 2 KO (1 male 1 female) or 3 WT (1 male 2 female) animals. Means (bars) and individual values (points) are shown. **d** Percentages of Ly49H⁺ or Ly49G2⁺ cells out of all splenic NK cells in *MAP8.B6.Ly49h* KO vs age and sex-matched WT animals. Ly49G2 is one of the protein isoforms of *B6.Ly49g* (alternative splicing)[91]. Means (bars*)* and individual values (points) are shown. Error bars: mean ± s.d. Two-tailed

unpaired Student's *t*-test. *P*-values are FDR-adjusted. For both KO and WT: *n* = 5 (3 male and 2 female mice). **e** Splenic NK cells from female KO vs WT animals in **d** were sorted. RNA-seq was performed on sorted cells. Normalization was performed using DESeq2 (Methods). **f** Viral load in the spleen and liver of mice 4 days after MCMV injection. Viral load was calculated as Log10 (1000 x copies of MCMV IE1/ copies of mouse Actin). Data were collected from 2 independent experiments. Experiment 1: *n* = 6 KO (4 male 2 female) and 7 WT (4 male 3 female) mice. Experiment 2: *n* = 5 KO (3 male 2 female) and 8 WT (4 male 4 female) mice. Means (bars) and individual values (points) are shown. Error bars: mean ± s.d. Two-tailed unpaired Student's *t*-test. *P*-values are FDR-adjusted. **g** Dotplot between the assembled contig and the WT B6 *Ly49* locus. **h** Schematic representation of *Ly49* enhancer choice. Source data are provided as a Source Data file.

differential usage of *MAP1/8* between inhibitory and activating *Ly49* genes. In CD8 T reg cells, compared to the *MAP1s* of inhibitory *Ly49* genes, the *MAP8s* of activating *B6.Ly49k, h*, and *n* showed notable decrease in chromatin accessibility, accompanied by the loss of H3K27ac marks (Supplementary Fig. 12a, b). Interestingly, such decrease in chromatin accessibility was not observed for the *MAP8* of the activating *B6.Ly49m*. Upon further investigation, we found that *B6.Ly49m* is in fact expressed in CD8 T reg cells (Supplementary Fig. 12c). Since *B6.Ly49m* has been pseudogenized and does not encode any proteins, it was missed by flow cytometry-based analyses in previous studies. It is likely that the NK-restricted expression pattern has been relaxed for *B6.Ly49m* due to a lack of evolutionary selection

against non-coding transcripts. Overall, our data showed that the differential CRE choice of inhibitory and activating *Ly49* genes is consistent with their differential expression pattern among NK cells and CD8 T reg cells.

Transcripts produced from *MAP1.B6.Ly49g* have been found in the mouse bone marrow, suggesting that *MAP1* elements have promoter activity in immature NK cells[17]. To test if *MAP8* elements could also have promoter activity in immature NK cells, we generated 2 nanoC-AGE libraries using FACS-isolated bone marrow NK cells (Supplementary Fig. 7c). Similar to splenic NK cells, *Pro2* still functions as the major *Ly49* TSS (Supplementary Fig. 13a, b). On the other hand, we captured abundant transcripts from *MAP1.B6.Ly49g*, some of which spliced into

downstream exons (Supplementary Fig. 13c), consistent with prior observations. As for *MAP8* elements, we observed weak nanoCAGE signals near *MAP8.B6.Ly49h*, and no evidence of splicing into downstream exons (Supplementary Fig. 13d). Although these could simply represent the noise of the nanoCAGE assay, they could also suggest weak *MAP8* promoter activity since the signals are reproducible across the 2 libraries. In conclusion, compared to *MAP1.B6.Ly49g*, we observed weak, if any, promoter activity at *MAP8* elements in bone marrow NK cells.

The *Ly49* locus is a sub-region of the *Nkc* complex, which contains 2 other families of NK receptors that are paralogous to the *Ly49* family (*Nkrp1* and *Nkg2*)[35]. However, no homologous regions ( $> = 100$ bp, $> = 60$ % sequence identity) of *MAP1/8* were detected in the *Nkc* complex outside of the *Ly49* locus (Supplementary Fig. 14a, Methods). *Nkrp1* genes are also known to encode either inhibitory or activating NK receptors[36]. Therefore, we investigated the evolutionary pattern of *Nkrp1* CREs to assess whether they exhibit any shared patterns with *MAP1/8s*. *Nkrp1c* (encoding NK1.1), the most studied *Nkrp1* gene, was used as the reference (Supplementary Fig. 14b). We did not observe any association between *Nkrp1* putative CRE (pCRE) conservation/accessibility and the inhibitory/activating functions of the encoded protein, which is partly due to the small number of gene family members available (Supplementary Fig. 14b).

## *Ly49* CRE choice likely resulted from lineage-specific CRE loss

To dissect the evolutionary history of *Ly49* CRE choice, we traced *Ly49* protein and *cis*-regulatory evolution using the single copy orthologous *Ly49* genes in squirrels, humans, cattle, and dogs (shcd *Ly49* genes). In mouse inhibitory Ly49 receptors, an immunoreceptor tyrosine-based inhibitory motif (ITIM) encoded in exon 2 transduces inhibitory signals[4]. This motif is conserved in shcd *Ly49* genes, with the exception of the pseudogene *human.Ly49* (Fig. 3a). In mouse activating Ly49 receptors, an arginine encoded in exon 3 allows association with a transmembrane signaling molecule that relays activating signals[4]. However, none of the shcd *Ly49* genes encode this arginine (Fig. 3a), suggesting that the inhibitory form is likely inherited from the *Ly49* ancestor, whereas the activating form is an evolutionary innovation in rodents, consistent with previous observations[4,37–39]. In fact, such derivation of activating NK receptors from an inhibitory ancestor was first observed in the human *KIR* family[40], which is the human counterpart of the mouse *Ly49* gene family.

The expression of *human.Ly49* is enriched in a cluster of T cells (cluster 5) in a human peripheral blood multiome dataset (Fig. 3b). The expression of *cattle.Ly49* and *dog.Ly49* is enriched in the spleen (Fig. 3c, d), among all organs examined. Using ATAC-seq data from human cluster 5 and cattle/dog spleen, we found a conserved enhancer-like element (*MAP0.5*) accessible in human, cattle, and dog (Fig. 3e, Supplementary Fig. 15a). Accessibility at *MAP0.5* is conserved in mouse inhibitory *Ly49* genes, but the summit has shifted upstream to *MAP1*, where the key transcription factor T-bet binds (Fig. 3e). By contrast, mouse *MAP8* is in a less conserved region (Supplementary Fig. 15b-c). Next, to investigate *Ly49* *cis*-regulatory evolution during its expansion in murids, we generated rat NK cell ATAC-seq data and constructed a *Ly49* phylogenetic tree using the sequences of introns 1 and 2 from mouse, rat, and golden hamster *Ly49* genes (Fig. 4a, Supplementary Fig. 15d-f, Methods). Mouse activating *Ly49* genes segregated into the same clade with a group of rat activating *Ly49* genes, suggesting that mutation events generated this activating *Ly49* clade in the common ancestor of mice and rats. All mouse activating *Ly49* genes are closely related to each other, suggesting recent amplifications in the mouse lineage. Their lack of accessibility at *MAP1s* likely reflects lineage-specific loss, since accessible *MAP1s* are present in various mouse and rat *Ly49* clades, including mouse and rat inhibitory *Ly49* genes. Notably, the closely related rat activating *Ly49* genes also have highly accessible *MAP1s* instead of

*MAP8s*, suggesting that the dependence of activating *Ly49* genes on *MAP8s* evolved after mice diverged from rats, highlighting the rapid and species-specific evolution of *cis*-regulatory elements upon recent gene duplications. Interestingly, the loss of *MAP1* accessibility in mouse activating *Ly49* genes is associated with the insertion of a ~ 2.5 Kb array of transposons between *MAP1* and the promoter, which might have disrupted their interactions (blue boxes in Fig. 4b). Similarly, mouse inhibitory *Ly49* genes were likely generated from recent mouse-specific amplifications. Their lack of accessibility at *MAP8s* also likely reflects lineage-specific loss, since accessible *MAP8s* were found upstream of not only activating *Ly49* genes, but also pseudogene *Ly49alpha* and distantly related paralogous gene fragment *Gm44182*. Furthermore, some rat *Ly49* genes (*rn7.Ly49i5/13*) appear to have both open *MAP1* and *MAP8*. Therefore, although we cannot completely rule out the separate gains of *MAP1/8* accessibility in multiple *Ly49* lineages, our data suggest that the association between mouse *Ly49* protein function and CRE choice is a consequence of the loss of regulatory activity at *MAP1/8* in mouse activating/inhibitory *Ly49* lineages, respectively (Fig. 4c).

## Concerted *cis*-regulation of different *Ly49* family members

Thus far, our data have defined a uniform epigenetic pattern for individual *Ly49* genes, namely a *cis*-regulatory unit consisted of one or more promoters and a proximal MAP. It is possible that the regulation of individual *Ly49* genes is confined to such units, and independent of the regulation of other *Ly49* genes. Indeed, the expression choice of *Ly49* family members has been described as independent of each other[14]. However, *Ly49* genes are clustered together within a small genomic space, raising the possibility that their *cis*-regulation might be orchestrated in a more coordinated fashion. To better examine this possibility, we assessed the co-expression pattern of B6 *Ly49* genes using a splenic NK cell scRNA-seq dataset[41]. The co-expression of multiple pairs of *Ly49* genes deviated significantly from independent expression-based expectations (Fig. 5a). For example, compared to NK cells not expressing *B6.Ly49m*, NK cells expressing *B6.Ly49m* have significantly higher odds of expressing *B6.Ly49a, d, j*, or *h*. To test if the non-independent expression pattern was reflective of the co-*cis*-regulation of these genes, we generated a knockout mouse model where the CRE (*MAP8*) of *B6.Ly49m* was removed (*MAP8.B6.Ly49m* KO; Fig. 5b, Supplementary Fig. 16a-d). In addition to the ablation of *B6.Ly49m* expression, *B6.Ly49j* expression was mostly abolished and *B6.Ly49a* expression was reduced by over 50 %. Modest downregulation of *B6.Ly49g* was also observed (Fig. 5c, d, Supplementary Fig. 16e). Interestingly, we did not observe a significant reduction of *B6.Ly49d* expression, despite the co-expression pattern between *B6.Ly49d* and *m*. This could be due to compensation from other *Ly49* enhancers in the locus, because our previously generated mouse model, ΔLy49-1, had large deletions encompassing both *B6.Ly49m* and *B6.Ly49h* and produced 80 % fewer Ly49D⁺ NK cells, despite that the gene body and *MAP1/8* elements of *B6.Ly49d* were intact[42]. Consistent with this, *B6.Ly49d* expression was moderately reduced in *MAP8.B6.Ly49h* KO mice (Fig. 5e). Furthermore, the proximal *MAP1/8* of both *B6.Ly49j* and *d* appeared to lack T-bet binding (Fig. 5c, black arrows). Together, our data demonstrate that different *Ly49* genes can be co-regulated through CRE-sharing, possibly facilitated by the decay of their proximal CREs' activities.

## 3D chromatin architecture of the *Ly49* locus

Members of a gene family could form characteristic 3D chromatin interaction patterns associated with their expression status, as extensively studied in the olfactory receptor gene family[43,44]. To explore the relationship between *Ly49* expression and the 3D chromatin architecture of the *Ly49* locus, we performed capture Hi-C on FACS-isolated Ly49D⁺ and Ly49D⁻ splenic NK cells from B6 mice. Based on gene composition and transcriptional activity, we partitioned the

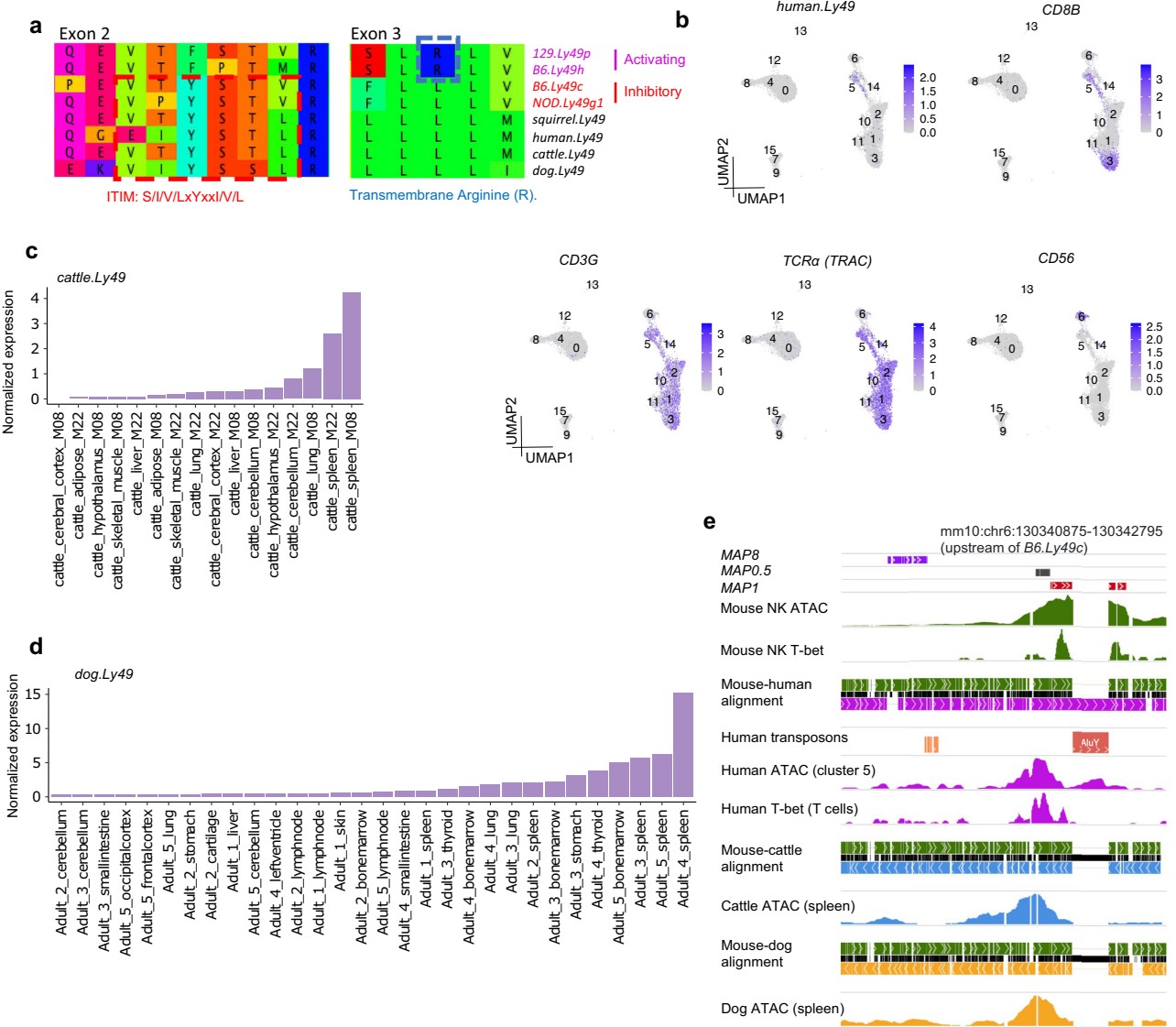

**Fig. 3 | *Ly49* coding sequence and *cis*-regulatory evolution across mammalian lineages. a** Amino acid sequence alignment of selected *Ly49* sequences from mice and the single copy *Ly49* sequences from squirrels, humans, cattle, and dogs. Amino acids were translated from DNA sequence alignment of the exons. **b–d** *Ly49* gene expression patterns in humans, cattle, and dogs. Human single cell multiome data was obtained from 10x Genomics public datasets and reprocessed in-house (Methods). *CD8B*, *CD3G*, and *TCRa* are markers for T cells. *CD56* marks human NK cells. Cattle RNA-seq data was obtained from GSE158430[92], re-analyzed using a custom pipeline (Methods), and normalized using cpm (counts per million).

Normalized dog gene expression values were directly obtained from BarkBase[93]. **e** WashU Comparative Epigenome Browser[94] view of the upstream region of *Ly49* genes. Data sources: Mouse NK ATAC: B6_Ly49Dp_ATAC_rep1 (this study, Supplementary Data 1); mouse NK cell T-bet ChIP-seq: GSM4314405[31]; human cluster 5 (**b**) ATAC-seq: 10x public multiome dataset (10 k PBMC from a healthy individual with granulocytes removed by sorting); human T-bet ChIP-seq: GSM776557[95]; cattle spleen ATAC-seq: GSM4799634[92]; dog spleen ATAC-seq: SRX5812510[93]. Multiple sequence alignment was generated using "mafft --auto".

neighborhood of *B6.Ly49d* into *cNK_Ly49* (expressed in conventional NK cells) and *ncNK_Ly49* (not expressed in conventional NK cells) regions (Methods). Overall, in Ly49D⁺ cells compared with Ly49D⁻ cells, *B6.Ly49d* exhibited increased interactions with the transcriptionally active *cNK_Ly49* region, and decreased interactions with the transcriptionally inactive *ncNK_Ly49* region (Supplementary Fig. 17a-b). Moreover, in Ly49D⁺ cells, *B6.Ly49d* formed local hotspots of chromatin contacts with 2 interaction partners located near *B6.Ly49m* and *B6.Ly49h*, respectively (Fig. 6a, b). These hotspots were not present in Ly49D⁻ cells. Notably, both interaction partners were deleted in the ΔLy49-1 mice (Fig. 6a, Supplementary Fig. 17c), which produced 80 % fewer Ly49D⁺ NK cells[42], suggesting that 3D chromatin interactions might play a part in the regulation of *B6.Ly49d* by distal CREs. Overall, these data indicate that the activation of *Ly49* expression is associated with 3D chromatin re-configuration.

Enhancers in the olfactory receptor gene family have been shown to form interchromosomal 3D enhancer hubs[43]. Similarly, our data showed that MAPs in the B6 *Ly49* locus appeared to form multiway interaction hotspots with each other in an NK cell-specific manner, suggesting that MAPs might form a local 3D hub (Fig. 6c–e, Supplementary Fig. 17d). Lastly, the *B6.Ly49d* promoter exhibited higher frequencies of chromatin interaction with most MAPs in Ly49D⁺ cells compared with Ly49D⁻ cells, suggesting that a *Ly49* promoter is only in close contact with the MAP hub when transcribed (Fig. 6f–h, Supplementary Fig. 17e). Together, these data describe the chromatin interaction patterns among *Ly49* genes associated with their expression.

## Discussion

Here we explored the *cis*-regulatory evolution of an immune gene family with recent and ongoing duplications. Although the proximal

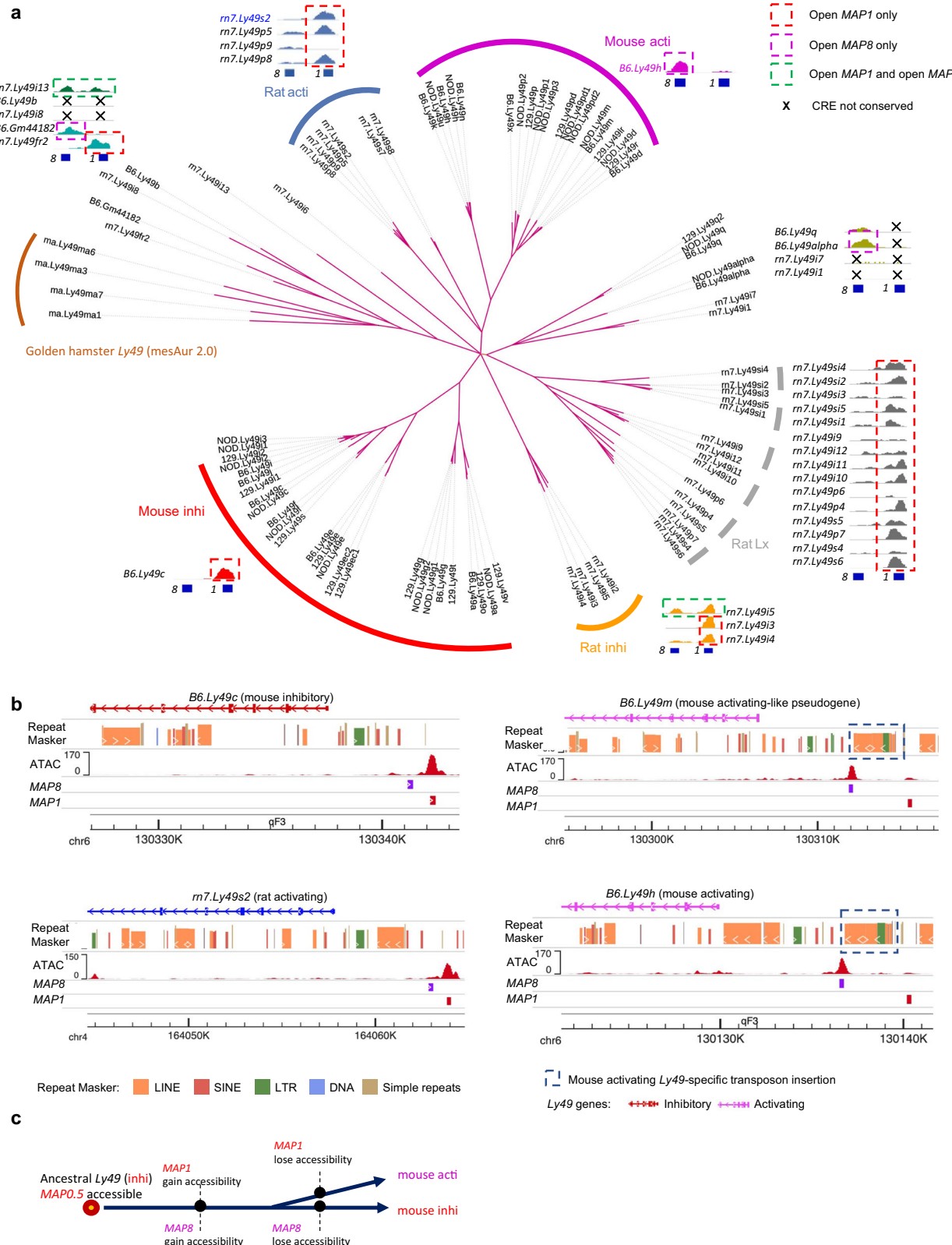

**Fig. 4 | Ly49 enhancer choice likely resulted from lineage-specific enhancer loss. a** Ly49 gene tree constructed from introns 1 and 2 using MrBayes (Methods). All branches shown have posterior probability > = 0.85. Branches with posterior probability < 0.85 have been deleted. rn7 indicate rat Ly49 genes. ATAC-seq signals (processed in the same way as Fig. 2a) on the coordinates of the Ly49 consensus sequence are shown for selected genes. A comprehensive view of the ATAC signals is presented in Supplementary Fig. 15d-e. Lx indicates the rat Ly49 clade characterized by the presence of an Lx transposon in intron 2. 1 and 8 represent MAP1 and MAP8, respectively. **b** Mouse activating Ly49 genes feature a ~ 2.5 Kb array of transposons inserted between MAP1 and promoter. **c** Model for the evolutionary mechanisms of Ly49 enhancer choice. Source data are provided as a Source Data file.

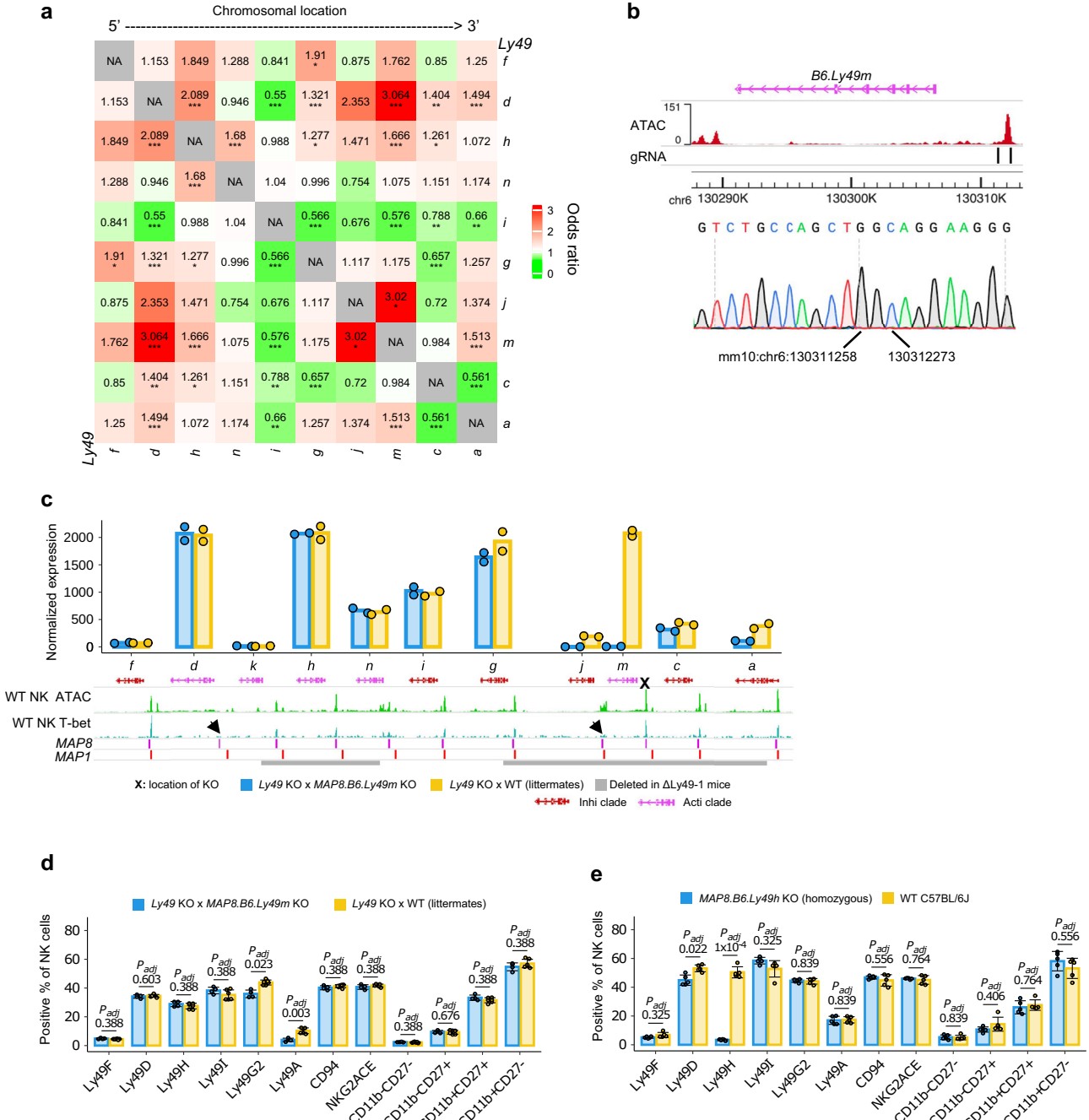

**Fig. 5 | Concerted *cis*-regulation of *Ly49* family members. a** *Ly49* co-expression pattern in splenic NK cells of B6 mice. Data source: GSE189807[41]. Entry (x, y) in the heatmap represents the odds ratio of gene *y* expression in cells expressing gene *x* relative to cells not expressing gene *x* (Methods). The heatmap is the average of 2 datasets (biological replicates). Asterisks indicate FDR-adjusted significance levels from Fisher's exact test. *FDR < 0.05; **FDR < 0.01; ***FDR < 0.001. Exact FDR values for each entry is available in Source Data. *n* = 4841 cells (sample 1) or 2632 cells (sample 2). **b** The exact genomic location of KO for the *MAP8.B6.Ly49m* KO allele. **c** The expression of *Ly49* genes (RNA-seq) from the *MAP8.B6.Ly49m* KO vs WT alleles. *n* = 2 independent biological replicates (female mice). Means (bars) and individual values (points) are shown. Gene expression was normalized by DESeq2. WT NK ATAC data: the B6_Ly49Dp_ATAC_rep1 sample generated in this study (Supplementary Data 1); WT NK T-bet ChIP-seq data: GSM4314405[31]. **d** The surface

expression (flow cytometry) of Ly49 receptors and other common NK surface markers in *Ly49* KO x *MAP8.B6.Ly49m* KO mice vs *Ly49* KO x *MAP8.B6.Ly49m* WT littermates. Ly49G2 is one of the protein isoforms of *B6.Ly49g* (alternative splicing)[91]. Means (bars) and individual values (points) are shown. Error bars: mean ± s.d. *P*-values were FDR-adjusted. Two-tailed unpaired Student's *t*-test. KO: *n* = 4 (2 male and 2 female mice); WT: *n* = 6 (3 male and 3 female mice). **e** The surface expression (measured by flow cytometry) of Ly49 receptors and other common NK surface markers in homozygous *MAP8.B6.Ly49h* KO mice vs age and sex-matched WT B6 animals. Ly49G2 is one of the protein isoforms of *B6.Ly49g* (alternative splicing)[91]. Means (bars) and individual values (points) are shown. Error bars: mean ± s.d. *P*-values were FDR-adjusted. Two-tailed unpaired Student's *t*-test. For both KO and WT: *n* = 3 male and 2 female mice. Source data are provided as a Source Data file.

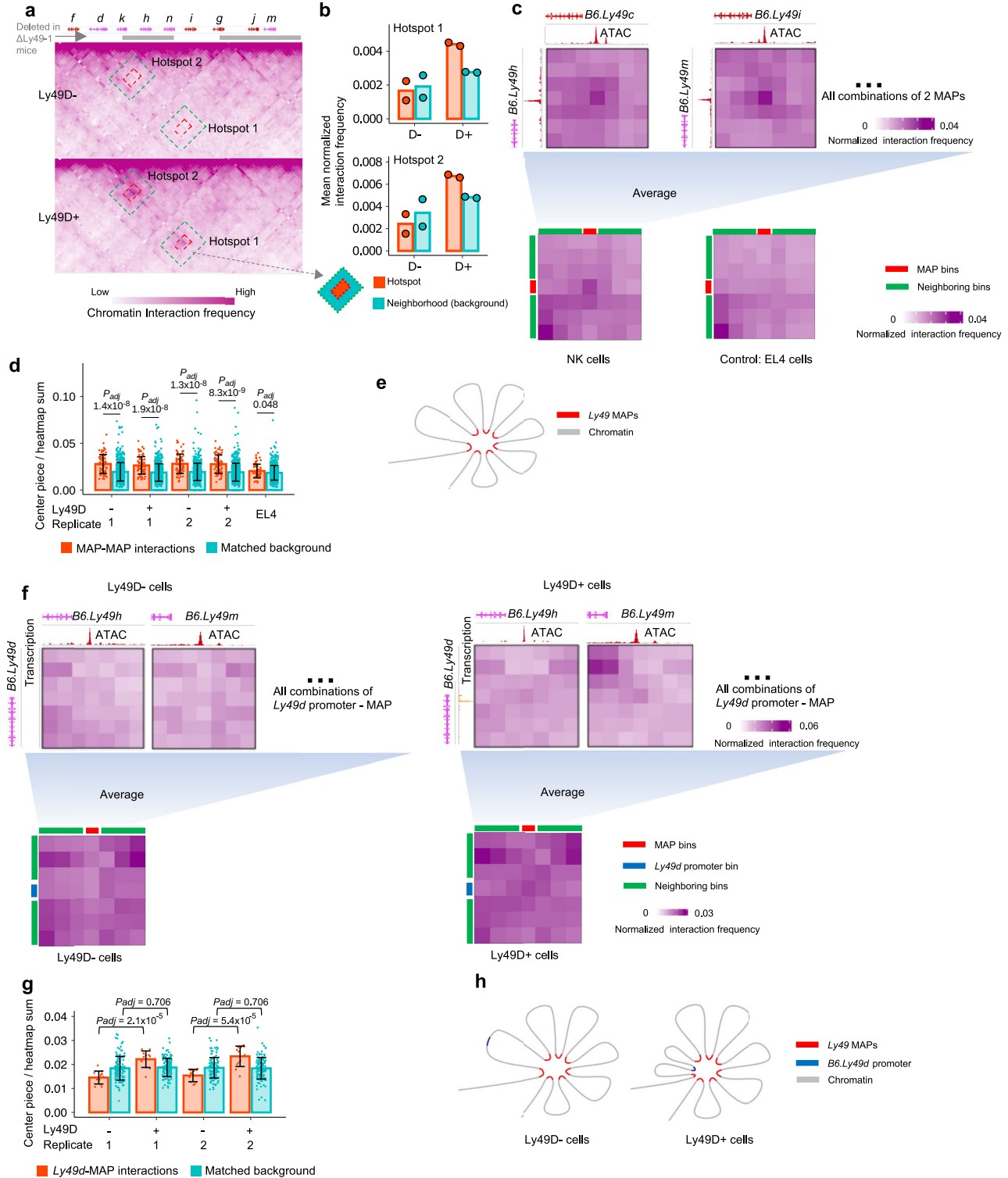

CREs were duplicated in the same segment as the gene, the mechanisms of *cis*-regulation were not simply cloned.

Our data show that both functional *MAP1s* and functional *MAP8s* were likely present in the common ancestor of mice and rats. Specifically in mice, however, we observed an association between *Ly49* protein function and CRE choice: Inhibitory *Ly49* genes are regulated by proximal *MAP1s*, whereas activating *Ly49* genes are regulated by proximal *MAP8s*. We focused on how this association might have evolved, as well as its functional consequence.

Although such association could be due to random chance, we noted the insertion of a ~2.5 Kb array of transposons between *MAP1* and promoter coincident with the loss of *MAP1* accessibility at mouse activating *Ly49* genes, which might have insulated *MAP1* from the gene, leading to the decay of *MAP1* and the dependence on *MAP8*. Furthermore, NKT cells[45] and CD8 T reg cells[34] express inhibitory Ly49 receptors but not activating Ly49 receptors, suggesting that differential CRE usage could underlie the broader expression of inhibitory *Ly49* genes versus the highly NK-restricted expression of

**Fig. 6 | The 3D chromatin architecture of the B6 *Ly49* locus. a** 3D chromatin interactions in the B6 *Ly49* locus at 5 Kb resolution. Red rectangles indicate the locations of the 2 hotspots in Ly49D⁺ cells. Their neighborhoods are defined as the regions between the green and red rectangles. **b** Quantification of the strengths of chromatin contacts between *B6.Ly49d* and its interacting partners for the 2 hotspots in (**a**). *Y* axes represent the mean VC-normalized interaction frequencies in a hotspot or its neighborhood. Means (bars) and individual values (points) are shown. *n* = 2 biologically independent samples. **c** MAP - MAP interaction frequencies (5 Kb resolution). 2 examples of MAP - MAP interaction profiles, and the average MAP - MAP interaction profiles calculated from all 66 combinations of 2 MAPs, with the one on the centromeric side as rows and the one on the telomeric side as columns. **d** Quantification of (**c**). Means (bars) and individual values (points) are shown. Each point represents a MAP - MAP interaction (*n* = 66) or a matched background interaction (*n* = 660). Error bars: mean ± s.d. Two-tailed unpaired

Student's *t*-test. *P*-values are FDR-adjusted. **e** Schema of the proposed 3D chromatin hub formed by *Ly49* MAPs. **f** *B6.Ly49d* promoter - MAP interaction frequency profiles (5 Kb resolution). 2 examples of *B6.Ly49d* promoter - MAP interaction profiles as a function of *B6.Ly49d* expression, and the average *B6.Ly49d* promoter - MAP interaction profiles calculated as the mean of individual profiles between the *B6.Ly49d* promoter and each MAP. **g** Quantification of (**f**). Means (bars) and individual values (points) are shown. Error bars: mean ± s.d. *P*-values are FDR-adjusted. Two-tailed unpaired Student's *t*-test. Two independent experiments are shown. For each experiment, each point represents the normalized chromatin contact frequency between *B6.Ly49d* promoter and one of the 12 MAPs (*n* = 12), or a matched background frequency (*n* = 120, 10 background for each MAP). **h** Schema of the proposed *B6.Ly49d* promoter - MAP hub interactions in Ly49D⁺ vs D⁻ cells. Source data are provided as a Source Data file.

activating *Ly49* genes, although we cannot rule out additional contributions from promoters and the local chromatin environment. In summary, the association between protein function and CRE choice could be evolutionarily advantageous.

Based on expression patterns, co-regulation has been reported as a general trend for tandem duplicate genes[46]. For olfactory receptors, paralogs in the same cluster share the same enhancers[43]. Similarly, for olfactory trace amine-associated receptors, the entire cluster is regulated by two enhancers[47]. However, it is generally unclear how the tight co-regulation has evolved. In the recently expanded *Ly49* family, some paralogs are regulated independently while others are co-regulated. In particular, the expression of *B6.Ly49j* is dependent on a proximal CRE of *B6.Ly49m*, and the expression of *B6.Ly49d* is largely dependent on distal CREs located in other *Ly49* genes. Importantly, the proximal MAPs of both *B6.Ly49d* and *j* have lost transcription factor binding, suggesting that the decay of proximal CREs could be an important evolutionary mechanism for the co-regulation of gene family members. We speculate that compared to closely spaced non-paralogous genes, CRE-sharing could happen more readily for young paralogs sharing similar temporal and spatial expression control, since their expression might be readily compensated by paralogous CREs. Furthermore, *B6.Ly49a* expression decreased by over 50% in *MAP8.B6.Ly49m* KO, suggesting *MAP8.B6.Ly49m* as the main contributor to *B6.Ly49a* expression. However, previous work has shown that when its proximal MAP was removed, *B6.Ly49a* expression was lost and not compensated by *MAP8.B6.Ly49m*[20], suggesting a complex, non-additive cooperation between proximal and distal CREs. Interestingly, despite the regulatory roles of *MAP8.B6.Ly49m*, *B6.Ly49m* itself has become a pseudogene due to a premature stop codon in exon 3[42], suggesting that in the birth-and-death[48] process of gene duplication, the survival of CREs could be independent of the survival of its target gene, due to the cross-regulation among different paralogs. In summary, we provide a snapshot into the early phase of a gene family's evolution from independent to concerted *cis*-regulation.

Although our study has charted the loss of CREs during *Ly49* duplications, the exact underlying mutations remain unclear. Murine NK cells are short-lived[49], and refractory to various transfection methods[50]. As a result, the validation of each candidate mutation would require the generation of a new mouse line. However, as in vitro transfection technologies continue to develop[51], we envision that the large-scale testing of candidate mutations in primary murine NK cells would be possible in future studies.

Our data highlight the rapid divergence and high complexity of *cis*-regulation during tandem gene duplications, demonstrating that the regulatory logic of one paralog might not fit others, even when they are closely related. In a human genetics setting, our data argue that systemic epigenetic mapping based on an accurate genome assembly could contribute to a better interpretation of non-coding variants at tandem duplicate loci.

## Methods
The research performed in this study complies with all relevant ethical regulations. Animal studies were approved by the Animal Studies Committee at Washington University School of Medicine under animal protocol 21-0090. Non-animal studies were approved by Environmental Health & Safety at Washington University in St. Louis under the approval number 10960.

### *Ly49* and genome sequences and annotations
All annotations generated in this study are available at https://github.com/ChangxuFan/Ly49evolution.

Mouse: For B6, the mm10 genome (analysis set, without fix/patch) was used to derive *Ly49* sequences and align sequencing reads. For the 129 strain, the GCA_001624185.1 assembly was used, with the *Ly49* region replaced by AY686474.1, AY686475.1, AY686476.1, AY686477.1, and AY686478.1. For the NOD strain, the GCA_001624675.1 assembly was used, with the *Ly49* region replaced by JH584264.1 and KK082443. *Ly49* annotations for the 129 and NOD strains are based on previous publications[23,24], and are available at our GitHub repo. For the BALB/cJ strain, the *Ly49* locus was assembled in[52], and kindly provided to us by Dr. Stephen Anderson.

Rat: *Ly49* sequences were taken from the rn7 genome (assembly: GCF_015227675.2, no fix/patch). Gene names and the assignment of activating vs inhibitory genes were adapted from a previous annotation of the rat *Ly49* locus[53]. *rn7.Ly49fr2* is a gene fragment not previously annotated and was found by BLAST. A gtf file containing all *Ly49* genes used is available at our GitHub repo. The Rat Lx clade was defined by the presence of an Lx transposon in intron 2. The rn7 genome was also used for rat NK ATAC-seq alignment.

Others: The following genome assemblies were used to derive *Ly49* sequences and map sequencing reads: golden hamster: mesAur 2.0; squirrel: speTri2; human: hg38; cattle: bosTau8; dog: canFam3. mesAur 2.0 was retrieved from NCBI assembly GCF_017639785.1. All other assemblies were retrieved from UCSC. Golden hamster *Ly49* genes were annotated based on the dotplot (generated using dotter[54]) between the *B6.Ly49c* locus and the putative *Ly49* locus of golden hamsters, defined as the region between *Olr1* and *Styk1*, according to the annotation retrieved from GSE200596[55]. Golden hamster *Ly49* genes (including pseudogenes) were named *Ly49ma1*, *Ly49ma2*, …, *Ly49ma7* based on their order on the contig. A copy of the golden hamster *Ly49* annotations is available at our GitHub repo. Annotations of single copy *Ly49* genes in other assemblies were retrieved from UCSC.

### Animals
C57BL/6 J (cat# 664), CB6F1/J (cat# 100007), and NOD/ShiLTJ (cat# 1976) mice were purchased from The Jackson Laboratory. 129S6/SvEvTac (cat# 129SVE) mice were purchased from Taconic Biosciences. BN/NHsdMcwi rats were purchased from Medical College of Wisconsin.

*Ly49* KO mice (Sytse J. Piersma, Shasha Li, Pamela Wong, Michael D. Bern, Jennifer Poursine-Laurent, Liping Yang, Diana L. Beckman, Bijal A. Parikh, and Wayne M. Yokoyama, manuscript in preparation) were generated through introducing 2 large deletions into the *Ly49* locus of B6 mice, one from *B6.Ly49q* to *B6.Ly49i*, the other from *B6.Ly49g* to *B6.Ly49a*. Therefore, *Ly49* KO mice do not express (on the protein level) any *Ly49* genes in the main *Ly49* cluster of the B6 mice.

For KO mouse models generated in this study, gRNAs flanking the indicated enhancers were designed using CRISPOR[56]. Candidate gRNAs with the highest specificity for the target region (compared to the entire chr6) were selected. For *MAP8.B6.Ly49h* KO, gRNAs (not including NGG) were: AGGUCUUAUGGAAUCAUCCA; AAGGCUAUCA-CAACCUCACC, while for *MAP8.B6.Ly49m* KO, gRNAs were: UCCAC-GAAUGACACCUCAGC; UGGCCCUGGCAUAUUCUGUA. Cas9 protein and gRNAs were ordered from IDT, and assembled Cas9 RNPs were delivered to single-cell embryos through electroporation. *MAP8.-B6.Ly49h* KO mice were genotyped using the primers TGCTTTTACC-CACTGGAACA; CTGCCCTTCTCTCCAACCTG with a Tm of 50 °C for 35 cycles using the Taq polymerase (M0488, New England Biolabs) according to manufacturer's instructions. *MAP8.B6.Ly49m* KO mice were genotyped using the primers ATGGGTAGTGGGGTGGAGAA and CTGGATCCAGCAATACCTCTCC with a Tm of 62 °C. For each KO mouse line, we backcrossed at least 3 separate founders harboring deletion alleles to *Ly49* KO mice to generate F1 mice that were either *Ly49* KO x *MAP8* KO or *Ly49* KO x *MAP8* WT, allowing phenotypic analysis of each founder. Data from one founder was shown for each KO mouse model as a representative. F1 mice were backcrossed to *Ly49* KO mice one more time to generate F2 animals, which were subsequently crossed to generate homozygous KO animals. All mouse models generated in this study are available upon request.

Animals were fed standard chow diet (LabDiet, PicoLab Rodent Diet 20, Cat# 53WU), in an ad libitum manner, and kept under an ambient temperature of 22 °C and 50-60% humidity with a 12 h dark/light cycle (6 am–6 pm: light; 6 pm–6 am: dark). Animals were euthanized using a 23.5 L $CO_2$ chamber with a flow rate of 50% of chamber volume per minute, according to AVMA Guidelines for the Euthanasia of Animals (2020 edition).

Mice were 8-16-week old at the start of experiments. Where possible, littermate controls were used, otherwise age- and sex-matched controls were used in all experiments. Female mice were used unless otherwise stated. Animal sex was determined by vendors, or determined according to the appearance of nipples and genital spacing. The phenotypes of KO animals generated in this study have been confirmed in both male and female mice. No sex-based analysis was performed, because sex-specific phenotypes in the regulation *Ly49* expression was not seen in this study or previous studies[18–20]. The sex for individual animals can be found in Source Data. This study was carried out in strict accordance with the recommendations in the Guide for the Care and Use of Laboratory Animals of the National Institutes of Health.

## Antibodies and markers for flow cytometry and cell sorting

All antibodies were used with 1:100 dilution, unless otherwise noted.

Antibodies for NK cells: The following antibodies and markers were used (anti-mouse unless otherwise indicated):

From BD Biosciences: CD49a (Ha31/8; PE; Cat# 562115), Ly49A (A1; Biotin; Cat# 557423), Ly49F (HBF-719; PE; Cat# 550987), NK1.1 (PK136, PE-Cy7; Cat# 552878), Streptavidin (PE; Cat# 554061);

From BioLegend: Ly49A (YE1/48.10.6; FITC; Cat# 116805), Ly49H (3D10; AF647; Cat# 144710), NK1.1 (PK136; BV650; Cat# 108735), Rat CD3 (1F4; PerCP-Cy5.5; Cat# 201417), Rat NKp46 (CD335) (WEN23; PE; Cat# 250803), Rat NKR-P1 (CD161) (3.2.3; APC; Cat# 205606), CD11b (M1/70; BV421; Cat# 101236; 1:200 dilution), Ly49C (4LO3311; AF647; custom conjugation via BioLegend; 1:50 dilution);

From Thermo Fisher: CD11b (M1/70; eF450; Cat# 48-0112-82), CD27 (LG.7F9; PE-Cy7, APC; Cat# 25-0271-82, 17-0271-82), CD3 (145-2C11; APC-eF780; Cat# 47-0031-82), CD19 (eBio1D3; APC-eF780; Cat# 47-0193-82), CD4 (RM4-5; APC-eF780; Cat# 47-0042-82), CD49b (DX5; eF450; Cat# 48-5971-82), CD8 (53-6.7; APC-eF780; Cat# 47-0081-82), CD94 (18d3; eF450; Cat# 48-0941-82), Ly49D (4D11; APC; Cat# 17-5782-82), Ly49E/F (CM4; PerCP; 46-5848-82), Ly49G2 (eBio4D11; FITC; Cat# 11-5781-82), Ly49H (3D10; FITC; Cat# 11-5886-82), Ly49I (YLI-90, FITC; Cat# 11-5895-85) NKG2A/C/E (20D5; FITC; Cat# 11-5896-85), NKG2AB6 (16a11; PerCP-eF710; Cat# 46-5897-82), NKp46 (29A1.4; PE-Cy7 1:25 dilution, PerCP-eF710, PE-eF610; Cat# 25-3351-82, 46-3351-82, 61-3351-82), TCRB (H597; APC-eF780; Cat# 47-5961-82), CD122 (TM-b1; PE; Cat# 12-1222-82), Viability (eF506; Cat# 65-0866; 1:500 dilution), Rat CD3 (eBioG4.18 (G4.18); Biotin; Cat# 13-0030-82; 1:250 dilution), Rat CD45R (B220) (HIS24; Biotin; Cat# 13-0460-82; 1:250 dilution), Ly49I (YLI-90; Biotin; Cat# MA5-28667);

From Jackson ImmunoResearch: anti-IgG3 (polyclonal; AF647; Cat# 115-605-209);

From Leinco: Ly49C (4LO3311, PE, Cat# L312).

Antibodies for CD8 T reg cells: The following biotinylated antibodies were used for negative selection of CD8 T reg cells (all from BioLegend): CD4 (clone RM4-5; Cat# 100508), CD19 (clone MB19-1; Cat# 101504), Ly-6G (clone 1A8; Cat# 127604), F4/80 (clone BM8; Cat# 123106), and CD14 (clone Sa14-2; Cat# 123306).

The following antibodies and markers were used for sorting CD8 T reg cells:

From BioLegend: CD4 (RM4-5; Alexa Fluor 700; Cat# 100536), CD8a (53-6.7; APC-Cy7; Cat# 100714), CD44 (IM7; PerCP; Cat# 103036; 1:50 dilution), Ly49C/F/I/H (14B11; PE; Cat# 108208), Ly49H (3D10; PE; Cat# 144706), Ly49D (4E5; PE; Cat# 138308), CD19 (6D5; PE-Cy7; Cat# 115520), CD3 (17A2; APC; Cat# 100236; 1:50 dilution);

From BD Biosciences: CD122 (TM-β1; BV421; Cat# 752988), Ly49F (HBF-719; PE; Cat# 550987), Ly49A (A1; PE; Cat# 557424);

From Invitrogen: Ly49I (YLI-90; PE; Cat# 12-5895-82); LIVE/DEAD Aqua Dead Cell Stain Kit (Cat# L34957);

From Miltenyi: Ly49G2 (REA1053; PE; Cat# 130-118-033), Ly49C/I (REAL296; PE; Cat# 130-118-940), Ly49E/F (REAL331; PE; Cat# 130-118-530; 1:50 dilution).

## Flow cytometry and cell sorting

Mouse NK cells (peripheral blood and spleen): Cheek-blood was collected in 50 mM EDTA-tubes, red blood cells were lysed with Tris-$NH_4Cl$ RBC lysis buffer and remaining cells were analyzed by flow cytometry. Single cell splenocyte solutions were obtained from spleens using cell strainers and treated with RBC lysis buffer. Where sorting was performed, NK cells were pre-enriched using the EasySep mouse NK cell isolation kit (Stemcell Technologies Cat# 19855 C), followed by surface staining with indicated antibodies in 2.4G2 hybridoma supernatant to block Fc receptors. NK cells were sorted as singlet CD4⁻CD8⁻TCRβ⁻CD19⁻NK1.1⁺NKp46⁺CD49a⁻CD49b⁺ (B6, CB6F1/J, and KO animals generated in this study) or singlet CD4⁻CD8⁻TCRβ⁻CD19⁻NKp46⁺CD49a⁻CD49b⁺ (129 and NOD, which do not encode *NK1.1* in their genomes). Cells were sorted on a FACSAria (BD Biosciences) into RPMI with 10% FBS for downstream library preparation. For phenotypic analysis without sorting, splenocyte or blood samples were stained with fixable viability dye (Thermo Fisher Scientific), continued by staining of cell surface molecules in 2.4G2 supernatant, followed by secondary antibodies and streptavidin. Where needed, samples were fixed in 1% paraformaldehyde until acquisition. Samples were acquired using FACSCanto (BD Biosciences) and analyzed using FlowJo (v10.4, BD Biosciences). NK cells were defined as singlet viability⁻CD4⁻CD8⁻TCRβ⁻CD19⁻NK1.1⁺NKp46⁺.

Mouse NK cells (bone marrow): To test the effect of *MAP8* deletion on *B6.Ly49h* expression, cells were collected from mouse tibiae through cutting off the distal end of the tibiae and centrifuging the

tibiae at 800 g for 3 min. RBC lysis and antibody staining were then performed similar to splenocytes. NK cells were identified as CD3⁻CD19⁻CD122⁺NK1.1⁺. NKp46, CD27, and CD11b were used to delineate NK cell maturation trajectory. To sort bone marrow NK cells for nanoCAGE, two strategies were tested: for library 1, NK cells were pre-enriched through negative selection similar to splenic NK cells, and sorted as CD3⁻CD122⁺NK1.1⁺CD49b(DX5)^low. For library 2, NK cells were sorted as CD3⁻CD122⁺NK1.1⁺CD49b(DX5)^high, without negative selection based enrichment.

Mouse NK cells (spleen, licensed vs unlicensed): NK cells were isolated as aforementioned for spleen, but identified using CD3⁻CD19⁻NK1.1⁺NKp46⁺. NKG2A, Ly49C, and Ly49I were used to identify licensed vs unlicensed NK cells.

Rat NK cells: Spleens were homogenized using cell strainers and treated with RBC lysis buffer to obtain single cell splenocyte solutions. To enrich for NK cells, splenocytes were stained with Biotinylated CD3 and CD45R (B220)-specific antibodies, followed by incubation with streptavidin RapidSpheres (Stemcell Technologies, component of Cat# 19855 C). CD3⁺ and CD45R⁺ cells were subsequently depleted using magnets. Pre-enriched NK cells were then stained with streptavidin-BV421 and anti-CD3, anti-NKp46, and anti-NKR-P1. Rat NK cells were defined as singlet CD3⁻CD45R⁻NKp46⁺NKRP1⁺ and sorted similar to mouse cells.

Mouse CD8 T reg cells: Single cell splenocyte solutions were obtained from spleens using cell strainers and treated with RBC lysis buffer. Next, negative selection was performed using Akadeum microbubbles (cat# 11110-000) and biotinylated antibodies listed above. Following surface staining, CD8 T reg cells were sorted as singlet live CD3⁺CD19⁻CD8a⁺CD44⁺CD122⁺Ly49⁺. Cells were sorted using FacsArialI (BD Biosciences).

## Murine cytomegalovirus (MCMV) infection

For virus studies, mice were infected i.p. with $5 \times 10^4$ PFU salivary gland WT1 MCMV as previously described[57]. Viral load was quantified in RNA-free organ DNA (extracted using Puregene DNA extraction kit (Qiagen, Cat# 1126462, 1045705, 1126468, 158136)) based on MCMV IE1 (forward: 5′-CCCTCTCCTAACTCTCCCTTT-3′; reverse: 5′-TGGTGCTC TTTTCCCGTG −3′; probe: 5′-TCTCTTGCCCCGTCCTGAAAACC-3′; IDT DNA) and mouse Actb (forward: 5′- AGCTCATTGTAGAAGGTGTGG-3′; reverse: 5′- GGTGGGAATGGGTCAGAAG-3′; probe: 5′-TTCAGGGTCAG-GATACCTCTCTTGCT-3′; IDT DNA) with plasmid standard curves using TAQman universal master mix II (Thermo Fisher, Cat# 4440048) on a StepOnePlus real time PCR system (Thermo Fisher Scientific) as previously described[58].

## nanoCAGE library construction and data analyses

For splenic NK samples: Splenic NK cells were FACS-sorted from B6, 129, and NOD mice. To confirm that our analysis pipeline does not map reads from one Ly49 gene to another, B6 cells were sorted into Ly49D⁺ vs Ly49D⁻ subsets, from which libraries were built separately. For all samples, total RNA was extracted using Quick-RNA Microprep Kit (Zymo Research, R1050). 20 ng of total RNA was used for each library. cDNA was generated from the RNA template using Template Switching RT Enzyme Mix (NEB, M0466) with hexamer priming. nanoCAGE libraries were then prepared following the previously described protocol[59]. For each of the two B6 samples (Ly49D⁺ and Ly49D⁻), we also generated one additional library where we used 250 ng total RNA as input, extracted mRNA, and further depleted truncated mRNA with Terminator 5′-Phosphate-Dependent Exonuclease (TER51020), which gave similar results (Supplementary Fig. 4). A summary of all libraries is available in Supplementary Data 1. For B6, the Ly49D⁺ sample without Terminator treatment was used to represent B6 NK cells, and all 4 samples were used to call nanoCAGE peaks and calculate the correlation between nanoCAGE and ATAC/ChIP-seq signals. Libraries were

sequenced on the Illumina NextSeq 500 platform. Each nanoCAGE library was sequenced to a depth of 8–10 million reads (2 x 75 bp).

For bone marrow NK samples: A low-input protocol was used due to the low cell number (library 1: ~ 2,000 cells; library 2: ~ 22,000 cells). Total RNA was extracted using Quick-RNA Microprep Kit (Zymo Research, R1050). cDNA was generated from the RNA template using NEBNext Single Cell/Low Input cDNA Synthesis & Amplification Module (NEB, E6421) with a Poly-T RT primer (20 µM):

TAGTCGAACTGAAGGTCTCCGAACCGCTCTTCC-GATCTTTTTTTTTTTTTTTTTTTTTTTVN and Template switching oligos (75 µM):

TAGTCGAACTGAAGGTCTCCAGCA(N1:25252525)(N1)(N1)(N1)(N1)(N1)(N1)(N1)TATArGrGrG

instead of the template switching oligos in the kit. (N1) represents one bp of the 9 bp UMI. 25252525 represents an equal mix of A/T/C/G. cDNA was then amplified using cDNA PCR primers (20 µM):

Forward: TAGTCGAACTGAAGGTCTCCAGCA;
Reverse: TAGTCGAACTGAAGGTCTCCGAACC

instead of the cDNA PCR primer in the kit. cDNA was quantified with Qubit and up to 0.5 ng of cDNA was applied for library preparation using Nextera XT library Prep Kit (Illumina, 15032350), following the previously described protocol[59]. Libraries were size-selected with double sided selection of 0.55x/0.95x Sera-Mag Select (Cytiva, 29343052) beads and were sequenced on the Element AVITI machine to ~ 20 million reads (2 × 75 bp).

Raw data were processed using a custom pipeline available at https://github.com/ChangxuFan/Ly49evolution: Template-switching oligonucleotide (TSO) and unique molecular identifiers (UMIs) were identified using TagDust2[60] (v2.33). Reads that do not have the expected architecture were discarded. All libraries contain > 90 % reads with the expected architecture; Sequencing adapters and TSO were trimmed using Cutadapt[61] (v1.18); STAR[62] (v2.5.4b) was used to align trimmed reads to the genome. B6 reads were aligned to mm10. 129 and NOD reads were aligned to custom-built genomes (Methods); PCR duplicates were removed according to UMI using UMI-tools (v1.0.1)[63]; alignments were further filtered to remove secondary alignments. Where indicated, multi-mapping reads were also removed (samtools view -f 0×2 -F 0×900 -q 255; samtools v1.7); a stringent secondary filter was applied to remove alignments satisfying any of the following criteria: a. template length > 10 kb. In the Ly49 locus, these alignments mostly contain cross-gene splicing; b. contain > 3 bp soft clipping; c. contain mismatches. This filter was feasible because all samples were from inbred animals and each sample was aligned to the reference Ly49 sequence built from exactly the same strain. Therefore, we do not expect any mismatches arising from genetic variations, and any deviation from the reference sequence likely represents errors in transcription, library prep, or data processing.

For B6 samples, nanoCAGE reads were converted to TSS signals using CAGEr (v1.30.3): CAGEr::getCTSS(correctSystematicG = FALSE, removeFirstG = FALSE)[64], followed by normalization using CAGEr::normalizeTagCount(ce, method = "powerLaw", fitInRange = c(5, 1000), alpha = 1.2, T = 10^6). Peaks were called based on normalized values using CAGEr::clusterCTSS (method = "distclu", threshold = 2, maxDist = 50, removeSingletons = T, keepSingletonsAbove = 4) in a 2 step fashion: a loose step where any pass-threshold TSS signals were used (nrPassThreshold = 1), and a strict step in which a TSS site is used only if all samples have pass-threshold signals (nrPassThreshold = 4) at that site. Only peaks detected in both loose and strict were used, with the exception of B6.Ly49d promoters, which were only required to pass the loose step, since they only have transcripts in Ly49D⁺ samples. The boundaries of accepted peaks were defined by the loose step, because the widths of peaks called from the strict step tended to be over-conservative. We further filtered the peaks to only retain those with > 70 % of reads showing the cap signature[65].

## Correlation between nanoCAGE and ATAC/ChIP-seq signals

Ly49D[+] vs Ly49D[-] (3 biological replicates each) ATAC-seq samples were used to match the nanoCAGE samples. For the correlation between nanoCAGE and ATAC signals at TSSs, we intersected nanoCAGE and ATAC peaks and correlated normalized nanoCAGE TSS counts (averaged across all 4 B6 samples mentioned in the nanoCAGE library construction and data analyses Methods section) in a nanoCAGE peak with the normalized number of ATAC Tn5 insertions in the corresponding ATAC-seq peak (averaged across 6 samples). nanoCAGE signals were normalized as mentioned above. ATAC signals were normalized to a total of 100000 across all used peaks. A similar approach was used to correlate nanoCAGE with histone signals. However, due to the variation in sample quality among public data sources, we extended each nanoCAGE peak upstream 1 Kb and downstream 3 Kb as the corresponding histone peak.

## ATAC-seq library construction and data analyses

All ATAC-seq data were generated according to the methods described in our previous publication[66]. Briefly, cells were harvested by centrifuging at 1000 g for 10 min at 4 °C. Supernatant was carefully aspirated and cell pellets were lysed in 100 μl of ATAC-seq RSB (10 mM Tris pH 7.4, 10 mM NaCl, 3 mM $MgCl_2$) containing 0.1 % NP40, 0.1 % Tween-20, and 0.01 % Digitonin by pipetting up and down and incubating on ice for 3 min. Next, 1 mL of ATAC-seq RSB containing 0.1 % Tween-20 was added and mixed with the lysis reaction. Nuclei were pelleted by centrifuging at 1000 g for 10 min at 4 °C. Supernatant was carefully removed, and nuclear pellets were resuspended in 25 μl 2x TD buffer (20 mM Tris pH 7.6, 10 mM $MgCl_2$, 20 % Dimethyl Formamide). 1ul of nuclei were checked using trypan blue under a microscope. Up to 50,000 nuclei were transferred to a tube with 2 × TD buffer filled up to 25 μL. 25 μl of transposition mix (2.5 μl transposase (100 nM final), 16.5 μl PBS, 0.5 μl 1 % digitonin, 0.5 μl 10 % Tween-20, and 5 μl $H_2O$) was then added to the nuclei. Transposition reactions were mixed and incubated at 37 °C for 30 min, and gently tapped every 10 min to mix. Reactions were cleaned up with Zymo DNA Clean and Concentrator 5 columns (Cat# D4014). ATAC-seq libraries were prepared by amplifying the DNA for 9-11cycles on a thermal cycler. The PCR reaction was purified with Sera-Mag Select (Cytiva, 29343052) beads using double size selection following the manufacture's protocol, in which 27.5 μl beads (0.55x sample volume) and 50 μl beads (1.5 x sample volume) were used based on 50 μl PCR reaction. ATAC-seq libraries were quantitated by Qubit assays. Paired-end ATAC-seq libraries were sequenced on an Illumina NextSeq 500 machine. 50 k cells were used for each library.

Mouse ATAC data were first processed using the AIAP[67] (v1.1) pipeline. As a part of the pipeline: locations of Tn5 insertion events were inferred from properly paired, non-PCR duplicate reads with MAPQ > = 10; each Tn5 insertion event was extended from insertion site up and down 75 bp to generate a 150 bp pseudo read; pseudo reads were piled up to generate an ATAC signal profile, which was normalized against sequencing depth. The resulting normalized ATAC signal profile was used in genome browser views unless otherwise noted. ATAC peaks were also called as a part of the AIAP pipeline. All MAPs in Supplementary Fig. 5e satisfy FDR < $10^{-20}$ in the B6_Ly49Hp_ATAC_rep1 sample.

To compare the Ly49 ATAC profiles of NK cells and CD8 T reg cells, we counted the number of aforementioned pseudo reads overlapping each MAP (MAP1 for inhibitory Ly49 genes, MAP8 for activating) to form a MAP-by-sample matrix. For NK cells, the Ly49A[+] and Ly49A[-] samples were used (2 samples each). Subsequently, counted values were normalized so that the total ATAC signals from all MAPs is constant across samples, and equal to one of the samples (rep2 of Ly49A[+] NK cells). This sample is chosen because its total MAP ATAC signals is the median of all samples.

For Ly49 consensus views of mouse and rat ATAC data, we fine-tuned the alignment using a custom alignment pipeline (https://github.com/ChangxuFan/snakeATAC/): adapters were removed using cutadapt (v1.18) with (--quality-cutoff = 15,10 --minimum-length = 36); reads were aligned to reference genomes using bowtie2 (--very-sensitive --xeq --dovetail --no-unal); only reads with perfect match to the reference were kept (sambamba view -f bam -F '[AS] = = 0'; sambamba v0.7.1). This was applied because we aligned ATAC data against Ly49 references assembled from exactly the same mouse/rat strains; where indicated, reads were filtered based on MAPQ (sambamba view -f bam -F 'mapping_quality > = 8 '); methylQA v0.2.1[68] (methylQA atac -X 38 -Q 1) was used to remove PCR duplicates, identify Tn5 insertion sites, and generate numeric ATAC signals, similar to the AIAP package.

NK cell ATAC-seq signals upstream of BALBc.Ly49l, the only activating Ly49 gene encoded in the BALB/c Ly49 locus, was obtained from GSM5492291[20], which was generated from NKG2A[+] NK cells of the F1 progeny of the B6 x BALB/c cross. Reads were aligned to a custom genome where the BALB/cJ Ly49 locus was added to the mm10 assembly, using the aforementioned pipeline. Only primary alignments with MAPQ > 1 were used (i.e. reads with low alignment scores in this locus or with better alignment scores in other loci are removed).

Public cattle and dog ATAC data were also aligned using this pipeline, but reads were not required to perfectly match the reference.

## Rescuing multimapping ATAC-seq reads

A schematic representation of the procedure is in Supplementary Fig. 6. First, reads were mapped to the Nkc locus (chr6:128534000-131409000 on mm10, containing the Ly49 locus) using Bowtie2[69] with high sensitivity parameters (v2.3.4.1; --very-sensitive --xeq --reorder --dovetail -a -p 3 --mm --no-unal -X 5000 --no-mixed). Next, alignment results were filtered based on the alignment score, but not MAPQ. We defined the normalized alignment score of a given alignment as nAS = AS:i / read_length. AS:i was calculated by Bowtie2. The highest possible value for AS:i is 0. Any mismatches, insertions, or deletions relative to the reference will lead to the reduction of AS:i. To estimate the distribution of nAS for a typical correct alignment, we aligned the same samples against the entire mm10 genome and assessed the nAS distribution of primary alignments across the genome (Supplementary Fig. 6b). Based on this distribution, we used −0.1 as the nAS cutoff for read rescue and removed any alignments with nAS < −0.1. Subsequently, for each Ly49 promoter, we removed reads that could map to other regions with a better alignment score (These were secondary alignments with MAPQ > 1). We note that this behavior is restricted to certain versions of Bowtie2, including v2.3.4.1 used in this study, but not the latest v2.5. Finally, for each Ly49 promoter region, we also included reads mapped there by the AIAP pipeline (using bwa mem). In summary, if a read could map to multiple regions within Nkc equally well (nAS > = −0.1), we assigned it to each of these regions; if a read could not map well to any regions within Nkc (nAS < −0.1), we discarded it; if a read could map well within Nkc (nAS > = −0.1) to >1 region, but a unique best mapping existed, we assigned it to the best mapping region. Notably, samples were generated using the inbred C67BL/6 J animals, from which the mm10 genome was derived. Therefore, we expect all correct alignments to have AS:i = 0 (i.e., no genetic variations expected). We set the cutoff at nAS > = −0.1 to allow for small amounts of errors in library prep and sequencing.

## Public ChIP-seq data analyses

A custom pipeline was used: Adapter sequences were detected through parsing the output of FastQC (v0.11.9); fastp (v0.20.0) was used to trim adapters[70]; Trimmed fastq files were aligned using Bowtie 2[69] (version 2.3.4.1; --sensitive --xeq --no-mixed --dovetail --mm --no-unal -k 5 -X 1000); For paired-end libraries, PCR duplicates were removed using Sambamba markdup; Alignment bam files were filtered for primary alignment only

and MAPQ > 8 using Sambamba view; ChIP signal profiles were generated through piling up pass-filter reads using bamCoverage (v3.5.0) from deepTools[71]. H3K4me3 peaks were called using macs2[72] (v2.1.1.20160309) with the --broad option and q value cutoff at 0.01.

## WGBS library preparation and data analyses

CB6F1/J (F1 progeny of C57BL/6 J and BALB/cJ) mice were used for bisulfite assays. The rationale is that the BALB/cJ genome does not encode *Ly49d* or *Ly49h*. Using CB6F1/J mice avoids the confounding factor that in Ly49D+ cells, *Ly49d* could be expressed from only one of the two alleles. The same applies to Ly49H. Genome DNA was extracted using Quick-DNA Microprep Kit (Zymo Research, R3020). DNA was sheared to produce 350 bp DNA fragments. 100 ng of DNA fragments were then bisulfite converted using EZ DNA Methylation-Gold Kit (Zymo, D5005) for subsequent WGBS library construction with xGen Methyl-Seq DNA Library Prep Kit (IDT, 10009860). WGBS libraries were sequenced on Illumina NovaSeq 6000 platform.

WGBS samples were processed using a custom pipeline publicly available at https://github.com/ChangxuFan/wgbs/tree/fanc. Briefly: trimgalore was used to remove adapters and low-quality bases; trimmed reads were aligned to the lambda genome using Bismark[73] to assess bisulfite conversion rates. All libraries have >99.5% conversion rates; trimmed reads were then aligned to the mm10 genome using Bismark, which also calculates the coverage and methylation level of each CpG across the genome; A custom script was used to convert Bismark outputs to methylC tracks, which were visualized using WashU Epigenome Browser.

## Amplicon based bisulfite sequencing and data analyses

Genome DNA was extracted using Quick-DNA Microprep Kit (Zymo Research, R3020). 500 ng of genomic DNA was bisulfite-converted using EZ DNA Methylation-Gold Kit (Zymo Research, D5005). The bisulfite-converted DNA was then amplified with target-specific primers using ZymoTaq PreMix (Zymo Research, E2003). Amplicon libraries were then generated using NEBNext Ultra II DNA Library Prep Kit (NEB, E7645). Libraries were sequenced on the Illumina NextSeq 500 platform. Primers used: forward: TTGATATAATATAGTTAAAAG GGTTTTTAGTA; reverse: TCATCTTCTCTATCTTCAAATATCATTTAT. Targeted region: mm10: chr6:130129675-130130083. Data were processed similar to WGBS. The percentage of reads with a given number of protected cytosines was counted with the help of the R function stackStringsFromBam() from the GenomicAlignments package.

## Targeted nanopore sequencing

High molecular weight DNA was extracted from ~5 million splenocytes of 1 *MAP8.B6.Ly49h* KO mouse, using Gentra Puregene Cell Kit (8 × 10$^8$) (catalog # 158767). 8 ug of DNA was then fragmented using Covaris g-TUBE (catalog 520079), which was spun at 4200 rpm to fragment the DNA to an average size of 50 Kb. The 7.7 ug DNA recovered after fragmentation was subsequently cleaned using Short Fragment Eliminator (Oxford Nanopore EXP-SFE001). The sequencing library was prepared using Ligation Sequencing Kit V14 (ONT SQK-LSK114) with 2 ug DNA as input, and 22 fmol of the library was run on a single ProMethION flow cell (FLO-PRO114M) for 42 h without reloading. This combination of ligation kit and flow cell is known to yield data at a modal accuracy above 99 %. Adaptive sampling was enabled to enrich for reads at the target region. Although the main *Ly49* locus is only about 700 Kb, we targeted a much larger region that contains the *Ly49* locus (chr6:110500000-140500000, about 1 % of the genome) to avoid a drastic decline of pore health due to the constant rejection of off-target reads.

Base-calling from raw sequencing data was perform using Guppy (v6.4.6) with the dna_r10.4.1_e8.2_400 bps_sup model. Base-called reads (fastq file) were aligned to the mouse genome (mm10) using minimap2[74] (v2.24, -ax map-ont). We then filtered the fastq file for reads mapped to the *Ly49* locus (chr6:129800001-130500000), irrespective of their mapping status (primary, secondary, or supplementary). These reads were used to perform de novo assembly of the *Ly49* locus using flye[75] (v2.9.2, --nano-hq -g 700 k). The dotplot between the assembled contig and the WT B6 *Ly49* locus (from mm10) was generated using the D-Genies[76] webtool. Sequence differences of the assembled contig compared to the WT locus was called using minimap2 (-x asm5 --cs = long -c) and paftools.js, which is distributed together with minimap2 (paftools.js call). The genomic locations of these sequence differences were intersected with the locations of repetitive elements of the genome, as determined by RepeatMasker. For each sequence difference, it is considered overlapping repetitive elements if it is entirely within repetitive elements, and otherwise considered as Non_repeat. We manually examined the sequence differences in the non_repeat category using IGV, which revealed that almost all of them were located at DNA homopolymers.

## Bulk RNA-seq library preparation and data analyses

Total RNA from sorted cells was extracted using Quick-RNA Microprep Kit (Zymo Research, R1050). 50–100 ng of total RNA was processed with NEBNext Poly(A) mRNA Magnetic Isolation Module (NEB, E7490) and NEBNext Ultra II Directional RNA Library Prep Kit (NEB, E7760) to generate mRNA-seq libraries, which were sequenced on Illumina NextSeq 500 platform.

A custom pipeline was used for the analyses of both in-house and public RNA-seq data (available at https://github.com/ChangxuFan/Ly49evolution): STAR (v2.5.4b) was used for alignment; sambamba markdup was used to remove PCR duplicates from paired-end libraries; reads with MAPQ < 30 were removed; subread (v2.0.0; feature-Counts -t exon --fracOverlap 0.5) was used to count the number of reads overlapping the exons of each gene. For cattle RNA-seq data, the count matrix of gene expression was normalized using edgeR::cpm()[77] (v3.26.8). DESeq2[78] (v1.26.0) was used to normalize and contrast: (1) the gene expression profiles of *Ly49* KO x *MAP8.B6.Ly49m* KO NK cells against *Ly49* KO x *MAP8.B6.Ly49m* WT NK cells; and (2) *MAP8.B6.Ly49h* KO (homozygous) NK cells against age- and sex-matched WT C57BL/6 J NK cells.

Public CD8 T reg cells RNA-seq data was normalized so that all samples have the same total gene expression level across all *Ly49* genes. The normalized sum is equal to the sample with GEO accession GSM3758133.

## Human PBMC single cell multiome data analyses

Fastq files of the 10x Genomics public dataset (10k PBMC from a healthy individual with granulocytes removed by sorting) was downloaded. Cellranger-arc (v2.0) was used for alignment. *KLRA1P* (*human.Ly49*) was added to the default hg38 reference (2020-A). For gene expression, reads aligned to introns were not counted. After alignment, Seurat[79] (v3.2.3) was used for gene expression data analyses. Briefly, cells were filtered based on "nFeature_RNA > = 500 & nFeature_RNA < = 2487 & percent.mito < 33". 2487 was found as an elbow point when nFeature_RNA was plotted against its rank. After this point nFeature_RNA increased rapidly. The cutoff of 33 for percent.mito (percentage of post-filter alignments mapped to mitochondria) was found similarly. Subsequently, NormalizeData, FindVariableFeatures, ScaleData, RunPCA, FindNeighbors, and RunUMAP were run with default parameters. Clusters were identified using FindClusters(resolution = 0.7). The ATAC-seq signal profile of cluster 5 was generated using getGroupBW(tileSize = 50) from the ArchR[80] (v1.0.1) package. 2 pixels of smoothing was used when visualizing the track on WashU Epigenome Browser.

## Capture Hi-C and 3D chromatin interactions

All Hi-C libraries were generated using the Arima Hi-C kit (part numbers A410231 and A410232), following the vendor's protocols. To capture the *Ly49* locus and its neighborhood, we designed oligos targeting chr6:128600000-130720000 (mm10) through https://earray.chem.agilent.com/suredesign, with 3x tiling density, moderately stringent repeat masking and boosting optimized for XT HS2. Capture was performed using SSEL XT HS2 DNA TE Kit (G9987A, Agilent), following the vendor's protocol (G9983-900000, Agilent). We generated 4 NK cell capture Hi-C samples (2 pairs of Ly49D$^+$ vs Ly49D$^-$ samples. Cells in each pair were sorted from the same animals), each comprised of 2 technical replicates. All samples were sequenced at 100–175 million reads (2 x 150 bp), generating 19–30 million Hi-C contacts, with an average on-target rate of 60 %. The EL4 capture Hi-C sample was processed similarly but sequenced to 8 million 2 x 75 bp reads. When the EL4 sample was contrasted with NK samples, NK samples were trimmed to 75 bp and randomly down-sampled to contain similar numbers of Hi-C contacts.

All capture Hi-C data were processed using the Juicer pipeline[81] (v1.6; -g mm10 -s Arima). Technical replicates were merged using the mega.sh script in Juicer. Processed data were visualized using WashU Epigenome Browser[82]. All capture Hi-C data were normalized using the VC method from the Juicer pipeline. The contact matrices based on MAPQ > 0 were used (inter.hic from the Juicer pipeline output files).

The *ncNK_Ly49* region (not expressed in conventional NK cells, mm10:chr6:129764477-130015298) contains *B6.Ly49e* and *B6.Ly49q*. Neither of them is expressed in conventional NK cells. The *cNK_Ly49* region (expressed in conventional NK cells, mm10:chr6:130105696-130391935) contains *Ly49* genes that are known to be mainly expressed in conventional NK cells (the cell type under study). The exact coordinates of these regions were expanded to be represented by 25 Kb Hi-C bins, based on which their interaction frequencies with *B6.Ly49d* were calculated.

MAP - MAP (M-M) interaction heatmaps and *Ly49d* promoter - MAP (P-M) interaction heatmaps were generated in R using the complexHeatmap[83] package. For each M-M or P-M interaction (fg), we selected 10 background interactions (bg) based on the following criteria: (1) genomic distance: the genomic distance between the 2 anchors of a bg interaction (dist.bg) must be similar to the distance between the 2 anchors of the fg (dist.fg): |dist.bg - dist.fg| ≤ 5 kb; (2) read coverage: for all interactions passing the genomic distance cutoff, we selected 10 interactions with read coverage at both anchors most similar to that of the fg, and used these 10 interactions as bg. The similarity of coverage was measured by the Euclidean distance between the fg and the bg in the 2D space formed by the read coverages of centromeric and telomeric anchors. The read coverage profile (VC normalization vector) of each sample was calculated by juicer and extracted using "juicer_tools.jar dump norm VC". Whenever multiple samples were used, the coverage profile of each sample was normalized to the same sum (100000). Then, the mean of these normalized coverage profiles was used to select the bg.

## *Ly49* consensus-based pileup views

The custom pipeline is available at https://github.com/ChangxuFan/Ly49evolution. Briefly, paralogous regions from different *Ly49* genes were aligned using MAFFT[84] (v7.427; --auto). Subsequently, R function Biostrings::consensusString(threshold = 0.5) (v2.54.0) was used to create the consensus sequence. Gaps in the consensus were removed. This process also created a correspondence map between each nucleotide position of individual *Ly49* sequences and the nucleotide positions on the consensus, which allowed us to project epigenetic signals from genomic coordinates to consensus coordinates, generating pileup views of nanoCAGE/ATAC/ChIP-seq signals.

For ATAC views, signals were generated using the snakeATAC pipeline (ATAC-seq library construction and data analyses Methods section). B6, 129, NOD, and rat data were represented by libraries B6_Ly49Dp_ATAC_rep1, 129_NK_ATAC_rep1, NOD_NK_ATAC_rep1, and rat_NK_ATAC_rep1, respectively (Supplementary Data 1). For ChIP-seq views, signals were generated by piling up entire reads. In ATAC or ChIP-seq views, signals in each track were normalized to the same sum over the plotted region on the consensus sequence. All tracks share the same y axis. When only very few reads are available in the entire region, these signals could give high normalized values and confound the visualization. Therefore, a uniform noise of 1.5 per bp of the consensus sequence was added to each track, which corresponds to the average genomic coverage of a 30 million paired-end 75 bp library for a 3 billion bp genome. BedGraph-like pileup views were generated using IGV.

For nanoCAGE views, bam files were further filtered to only retain reads with the cap signature (transcripts beginning with at least one unencoded G)[65]. For nanoCAGE views from BM NK cells, we further identified alignments in close proximity (within 100,000 records from each other in a position-sorted bam file) with the same UMI, and only kept the first alignment for each UMI. This was performed as a precaution because bone marrow libraries were constructed from low cell numbers, and are more susceptible to artifacts caused by the PCR amplification of cDNA. CAGEr[64] was then used to convert bam files to TSS signals in the bedGraph format, which was subsequently projected onto the consensus coordinates and visualized using IGV. Only TSS signals consistent with the orientation (negative strand) of *Ly49* transcripts were kept (very few reads were filtered out this way). The y axis of each track is independent: the ymax of each track was set to the maximum TSS signal of that track, since the question of interest is the distribution of TSS signals, not their absolute values. We only included protein-coding *Ly49* genes with at least 50 read-pairs mapped to exons, after all filters were applied except the MAPQ filter. Applying MAPQ filters could bias TSS signal profiles. Consider a gene with major TSS at site A and minor TSS at site B. If site A has low mappability, filtering by MAPQ will remove reads overlapping site A and bias the summit of TSS signals towards site B. However, without MAPQ filters, closely related paralogs cannot be distinguished. We present both versions. The effect of MAPQ filters is illustrated by *NOD.Ly49p1* and *NOD.Ly49p3* (Supplementary Fig. 3a, c). Their sequences are almost identical. Filtering by MAPQ led to the removal of reads from *Pro2*, resulting in higher relative signal values at *Pro3*. The heatmap view of TSS signals were generated using ggplot2 (v3.3.3) in R. For B6, the B6_Ly49Dp_mRNA (Supplementary Data 1) sample was used to maintain consistency with the other two strains.

## Runx3 motif scanning in *MAP1s* and *MAP8s*

For the proximal *MAP1s* of mouse inhibitory *Ly49* genes and the proximal *MAP8s* of the mouse activating *Ly49* genes, we first defined highly conserved sequence elements as those >= 10 bp with >= 90% of the paralogs having identical nucleotides at each position. Then we scanned these elements for the presence of the Runx3 binding motif (MA0684.1 from JASPAR[85]) using FIMO[86] with the *p*-value cutoff at 0.01. The actual *p*-values of discovered motifs are indicated in figures.

## Sequence conservation of mouse, human, cattle, and dog *Ly49* genes

VISTA[87] (v1.4.26) was used to calculate the percentage identity among *B6.Ly49c*, *B6.Ly49h*, *squirrel.Ly49*, *human.Ly49*, *cattle.Ly49*, and *dog.Ly49*, based on the pairwise alignments generated by mafft --auto. The default sliding window size (100 bp) was used. Regions longer than 100 bp with percent identity >70 were highlighted. The phastCons signal track and the phastCons conserved elements track were downloaded from UCSC table browser[88] (hg38/phastCons100way and hg38/phastConsElements100way).

### *Ly49* gene tree construction

It is known that exons 1–3 and exons 4–7 of murine *Ly49* genes evolved under 2 distinct patterns[22,23]. Since the inhibitory/activating activities of *Ly49* genes are encoded within exons 2 and 3, sequences from introns 1 and 2 were used to construct the *Ly49* gene tree. We first removed sequences within 200 bp of exons. We then aligned *Ly49* paralogs from golden hamsters, mice, and rats using "mafft --auto". We additionally removed the first 271 bp from intron 2 due to poor alignment. The resulting regions, when mapped to the coordinates of *B6.Ly49c*, corresponds to mm10:chr6:130336017-130337271 (intron 1) and mm10:chr6:130334504-130335217 (intron 2). Lastly, we removed positions that are gaps in > 50 % of the sequences used, leading to a 1314 bp alignment for intron 1 and a 749 bp alignment for intron 2. MrBayes[89] (v3.2) was used to construct the *Ly49* gene tree. Introns 1 and 2 were treated as two separate partitions. For each partition, a GTR + I + Γ model was fitted. Parameters for 2 partitions were unlinked. Mcmc was run for 1 million generations with default priors. The following genes were excluded due to the absence or incompleteness of sequences at intron 1 and/or intron 2: *129.Ly49alpha, 129.Ly49ui, 129.Ly49u, 129.Ly49q3, NOD.Ly49x, NOD.Ly49alpha2, rn7.Ly49p1, rn7.Ly49p2, rn7.Ly49p3*, and *rn7.Ly49fr1*.

### *Ly49* co-expression patterns

The fastq files of splenic samples (2 biological replicates) from Lopes et al.[41] were downloaded from NCBI SRA. Cellranger (v6.0) was used to generate the gene-by-cell expression matrices. Protein coding genes from Gencode V24 were used for gene annotation. Activating receptor-like pseudogenes (*B6.Ly49m*, *n*, and *k*) were manually added. Reads aligned to introns were not counted. We modified the default cellranger count output to remove any reads with the MM tag. Reads with the MM tag are multi-mapping as determined by STAR, but their MAPQ values were modified to 255 (uniquely mapped) by cellranger due to various reasons. We removed these reads to only retain reads determined as uniquely mapped by STAR. The resulting gene-by-cell matrices were imported into R and further processed with Seurat, similar to the analysis of Human PBMC single cell multiome data. Briefly, cells were filtered based on "nFeature_RNA > = 1000 & nFeature_RNA < = 2500 & percent.mito < elbow_point". The 2 splenic samples were then merged, and cells were clustered as mentioned above with resolution = 0.5. We then removed 4 small clusters (~3.8% of all cells) that might represent non-NK cell types.

*Ly49* co-expression patterns were calculated based on binarized gene-by-cell matrices, based on the previous observation that in a given NK cell, a *Ly49* gene is either fully expressed or unexpressed, leading to the bimodal distribution in flow cytometry[20]. Cells from the two splenic samples were analyzed separately. For each sample, we partitioned all NK cells into the following contingency table for each pair of *Ly49* genes $(x, y)$, where "+" indicates at least one pass-filter read from the cell was mapped to the gene, and "-" indicates that no pass-filter reads from that cell was mapped to the gene:

|      | $x-$ | $x+$ |
|------|------|------|
| $y-$ | $a$  | $c$  |
| $y+$ | $b$  | $d$  |

$a$, $b$, $c$, and $d$ represent the number of cells in each category. The odds ratio of gene $y$'s expression in $x+$ vs $x-$ cells was calculated as $(d/c)/(b/a)$, and presented in the entry $(x, y)$ of the heatmap. *P*-values were calculated using Fisher's exact test in R (fisher.test()) and adjusted using the FDR method (p.adjust(method = "fdr")). Each entry of the presented heatmap is the mean of the two samples, and its significance level is the less significant of the two.

### *Nkrp1* family analyses

To determine if there are *MAP1/8* homologs in the *Nkrp1* locus, UGENE was used to generate dot plots between the *Nkc* and the *MAP1* or *MAP8* consensus sequence. Homology was defined as regions ≥100 bp with ≥60 % sequence identity. Both direct and inverted homologs were searched. The default algorithm was used.

To study the *cis*-regulatory evolution of the *Nkrp1* family, we defined the gene body ± 10 Kb region of each gene as its *cis*-regulatory region. The *cis*-regulatory region of each gene was aligned in a pairwise manner to that of *Nkrp1c* using "mafft --auto". Percent identity was calculated using VISTA as mentioned above. The NK cell ATAC-seq profile of each *Nkrp1* gene was projected onto the *Nkrp1c* locus based on its alignment against the *Nkrp1c* locus.

### Statistical analysis

R was used to perform all statistical analyses. Two-tailed Student's *t*-tests (*t*.test() in R) were used unless otherwise noted. All error bars represent mean ± standard deviation (s.d).

### Reporting summary

Further information on research design is available in the Nature Portfolio Reporting Summary linked to this article.

## Data availability

All sequencing data generated in this study have been deposited in GEO DataSets under the accession code GSE226502, and are publicly available. Other datasets used in this study are also publicly available at GEO DataSets: Mouse CD8 T reg cells: RNA-seq: GSE133364; H3K27ac: GSM1876376. Mouse NK ChIP-seq: H3K4me3: GSM4314407 and GSM4314396; H3K27ac: GSM4314409; p300: GSM2056372; T-bet: GSM4314405; Runx3: GSM1214531. Human T-bet ChIP-seq: GSM776557. Cattle spleen ATAC: GSM4799634. Dog spleen ATAC: SRX5812510. Cattle RNA-seq: GSE158430. Dog RNA-seq: barkbase [https://data.broadinstitute.org/barkbase/RNA-seq/]. Golden hamster genome annotation: GSE200596. ATAC-seq from NKG2A+ NK cells of the F1 progeny of the B6 x BALB/c cross: GSM5492291. Mouse iNK ATAC: GSM2056300. Mouse pDC ATAC: GSM2692341. Mouse splenic NK single-cell RNA-seq: GSE189807. Human peripheral blood single cell multiome data is publicly available from 10x website [https://www.10xgenomics.com/datasets/pbmc-from-a-healthy-donor-granulocytes-removed-through-cell-sorting-10-k-1-standard-2-0-0]. Flow cytometry data are available in the Source Data file. Source data are provided with this paper.

## Code availability

Code for bioinformatic analyses in this paper is available at Github [https://github.com/ChangxuFan/Ly49evolution] and Zenodo[90] [https://doi.org/10.5281/zenodo.10998685].

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

## Acknowledgements

We would like to thank Jessica Hoisington-López and MariaLynn Jaeger from The Edison Family Center for Genome Sciences & Systems Biology

(CGSSB) at Wash U for assistance with sequencing. We thank Zi Teng Wang, Xiaoxia Cui, and Jane Kouranova from The Genome Engineering & Stem Cell Center (GESC) at Wash U for the design and validation of gRNAs and sequencing primers. We thank Mike White and his team at the Transgenic, Knockout and Micro-Injection Core at Wash U for generating the KO mice. We thank Catrina Fronick at McDonnell Genome Institute for nanopore experiments. We thank Shasha Li, Molly Keppel, and the Siteman Flow Cytometry Core at Wash U for assistance in FACS. We also thank Stephen Anderson at National Cancer Institute for the assembly of the *Ly49* locus in 129, BALBc, and NOD mice. C.F., X.X., H.S., M.N.K.C., and T.W. are supported by National Institutes of Health grants R01HG007175 (T.W.), U24ES026699 (T.W.), U01HG009391 (T.W.), and U41HG010972 (T.W.). J.P., B.A.P., J.Y., S.J.P. and W.M.Y. are supported by National Institutes of Health grant R01AI129545 (W.M.Y.). N.S. is supported by National Multiple Sclerosis Society Career Transition Award TA-1804-30600 (N.S.). S.J.H.M. is supported by Department of Neurology start-up funds (N.S.) and Medical Scientist Training Program at Washington University in St. Louis.

## Author contributions

C.F. conceived the study, performed all bioinformatic analyses, as well as flow cytometry and cell sorting for bone marrow NK cells and licensed vs unlicensed splenic NK cells. S.J.P., W.M.Y., and T.W. supervised the study. S.J.P. performed infection experiments and flow cytometry, including cell sorting. X.X. generated all sequencing libraries. J.P. maintained the animal colony. H.S. genotyped animals. B.A.P. generated the *Ly49* KO mice. J.Y. validated gRNAs. M.N.K.C piloted capture Hi-C. S.J.H.M. sorted CD8 T reg cells, under the supervision of N.S. C.F. wrote the first draft of the manuscript with input from all authors. S.J.P., W.M.Y., and T.W. revised the manuscript.

## Competing interests

The authors have no competing interests.
