## [Peer Review File · Nature Communications]

Cis-regulatory evolution of the recently expanded Ly49 gene familyEditorial Note: This manuscript has been previously reviewed at another journal that is not operating a transparent peer review scheme. This document only contains reviewer comments and rebuttal letters for versions considered at *Nature Communications*.

REVIEWERS' COMMENTS

Reviewer #1 (Remarks to the Author):

I am satisfied with the author's attempts to answer my questions especially with respect to the Nkrp1 genes, Ly49 expression by T cells, immature NK cells, and the Balb/c Ly49L gene.

The lack of a functional MAP8 site in the canonical activating Ly49, Ly49D, shows that MAP8 is not critical for all activating Ly49 expression in NK cells. This should be clearly stated in the manuscript.

Reviewer #2 (Remarks to the Author):

The revised Fan et al manuscript has thoroughly addressed all of my concerns with additional data and modifications to the text.

Reviewer #3 (Remarks to the Author):

The revised report from Fan and colleagues establishes an evolutionary basis of complex cis-regulatory control of the family of Ly49 receptors in mice that affect NK functionality and pathogen control. Their work advances understanding of Ly49 gene expression, the adaptations through evolution that have given rise to discrete regulatory mechanisms for activating and inhibitory receptors and a blueprint to investigate other notable multigenic NK receptor families. The authors have thoroughly responded to and addressed my prior comments and concerns. The findings are well presented, and I agree that the conclusions are justified. I have no additional concerns.

Reviewer #1 (Remarks to the Author):

I am satisfied with the author's attempts to answer my questions especially with respect to the Nkrp1 genes, Ly49 expression by T cells, immature NK cells, and the Balb/c Ly49L gene.

5 The lack of a functional MAP8 site in the canonical activating Ly49, Ly49D, shows that MAP8 is not critical for all activating Ly49 expression in NK cells. This should be clearly stated in the manuscript.

We agree that proximal *MAP1* and *MAP8* elements cannot explain the expression of all *Ly49* genes. This has been clearly stated in the revised manuscript in line with the reviewer's suggestion, and provided here for convenience (added text underlined):

10 Together, these data strongly suggest that *MAP1s* and *MAP8s* act as enhancers regulating the expression of inhibitory and activating *Ly49* genes, respectively (Fig. 2h). Interestingly, *B6.Ly49d*, despite being highly expressed in NK cells, do not have accessible proximal *MAP1* or *MAP8*, suggesting that not all *Ly49* genes are regulated by proximal *MAP1s* or *MAP8s*. At the DNA sequence level, *MAP1* and *MAP8* paralogs are both conserved upstream of each *Ly49* gene, with very few exceptions.

15

Reviewer #2 (Remarks to the Author):

The revised Fan et al manuscript has thoroughly addressed all of my concerns with additional data and modifications to the text.

20 We highly appreciate the reviewer's kind remarks.

Reviewer #3 (Remarks to the Author):

25 The revised report from Fan and colleagues establishes an evolutionary basis of complex cis-regulatory control of the family of Ly49 receptors in mice that affect NK functionality and pathogen control. Their work advances understanding of Ly49 gene expression, the adaptations through evolution that have given rise to discrete regulatory mechanisms for activating and inhibitory receptors and a blueprint to investigate other notable multigenic NK receptor families. The authors have thoroughly responded to and addressed my prior comments and concerns. The findings are well presented, and I agree that the conclusions are justified. I have no additional concerns.

30

We highly appreciate the reviewer's kind remarks.